# PASS: Path-selective State Space Model for Event-based Recognition

**Jiazhou Zhou**
AI Thrust, HKUST(GZ) *
International Digital Economy Academy
jzhou297@connect.hkust-gz.edu.cn

**Kanghao Chen**
AI Thrust, HKUST(GZ)
kchen879@connect.hkust-gz.edu.cn

**Lei Zhang**
International Digital Economy Academy
leizhang@idea.edu.cn

**Lin Wang**[†]
School of Electrical and Electronic Engineering,
Nanyang Technological University
linwang@ntu.edu.sg

## Abstract

Event cameras are bio-inspired sensors that capture intensity changes asynchronously with distinct advantages, such as high temporal resolution. Existing methods for event-based object/action recognition predominantly sample and convert event representation at every fixed temporal interval (or frequency). However, they are constrained to processing a limited number of event lengths and show poor frequency generalization, thus not fully leveraging the event's high temporal resolution. In this paper, we present our PASS framework, exhibiting superior capacity for spatiotemporal event modeling towards a larger number of event lengths and generalization across varying inference temporal frequencies. Our key insight is to learn adaptively encoded event features via the state space models (SSMs), whose linear complexity and generalization on input frequency make them ideal for processing high temporal resolution events. Specifically, we propose a Path-selective Event Aggregation and Scan (PEAS) module to encode events into features with fixed dimensions by adaptively scanning and selecting aggregated event presentations. On top of it, we introduce a novel Multi-faceted Selection Guiding (MSG) loss to minimize the randomness and redundancy of the encoded features during the PEAS selection process. Our method outperforms prior methods on five public datasets and shows strong generalization across varying inference frequencies with less accuracy drop (ours -8.62% *v.s.* -20.69% for the baseline). Overall, PASS exhibits strong long spatiotemporal modeling for a broader distribution of event length $(1-10^9)$, precise temporal perception, and generalization for real-world scenarios.

## 1 Introduction

Event cameras are bio-inspired sensors that trigger signals when the relative intensity change exceeds a threshold, adapting to scene brightness, motion, and texture. Compared with standard cameras, event cameras output asynchronous event streams, instead of fixed frame rates. They offer distinct advantages, such as high dynamic range, microsecond temporal resolution, and low latency [20, 15, 73, 6]. Due to these merits, event cameras have been applied to address various vision tasks, such as object/action recognition [9, 4, 31, 74, 76, 77, 54, 7, 17, 16, 35, 56, 55, 48, 1].

---

*Project page: https://github.com/jiazhou-garland/PASS_Homepage.
†Corresponding Author.

39th Conference on Neural Information Processing Systems (NeurIPS 2025).

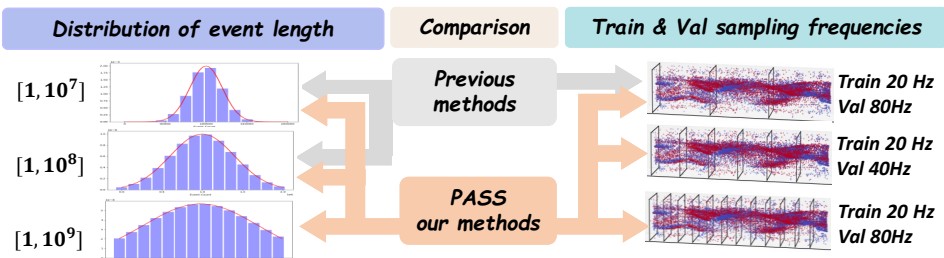

Figure 1: Compared to previous event-based recognition methods limited to a narrow distribution of event length and poor temporal frequency generalization, our method, PASS, advances spatial-temporal event modeling across a broader distribution of event length ranging from 1 to $10^9$ and demonstrates superior temporal frequency generalization.

The spatiotemporal richness of events introduces complexities in data processing and necessitates models that can efficiently process and interpret them. To address this problem, existing methods predominantly sample and aggregate them at every fixed temporal interval, *i.e.*, frequency. In this way, the raw stream can be converted into different event representations [75, 80, 2, 54, 66, 44, 32, 58]. In general, existing methods mainly follow two representative model structures: (a) step-by-step structure models [66, 69, 77, 75, 74, 30, 16] and (b) recurrent structure models [54, 79]. The former processes all time-step event frames in parallel, employing local-range and long-range temporal modeling sequentially, as shown in Fig. 2 (a). By contrast, the latter process event frames sequentially at each time step, updating a memory feature that affects the next input, as illustrated in Fig. 2 (b).

However, current models face two pivotal challenges, as shown in Fig. 1. 1) Limited distribution of event length. Event cameras offer high temporal resolution, naturally generating dense event sequences. This necessitates models effectively processing events across a broad distribution of event length, especially for high-speed scenarios or long-duration event streams [80]. However, current event-based recognition datasets [8, 21, 47] are restricted to a limited number of event lengths ($10^6 - 10^7$) (see appendix for existing dataset summary) and face computational bottlenecks for large event lengths due to quadratic attention complexity, thereby constraining exploration of spatiotemporal relationships across a broad distribution of event lengths. 2) Limited inference frequency generalization. While event-based cameras offer high temporal resolution beneficial for recognizing objects and actions in high-speed, dynamic visual scenarios [80], current recognition models significantly degrade when inference frequencies differ from the training one, thereby limiting the full potential of these high-resolution event streams. For example, as shown in Fig. 5 (b), the model trained at 60 Hz with existing event sampling strategies demonstrates poor generalization, with performance dropping up to 20.69% when evaluated at 20 Hz and 100 Hz sampling frequencies.

Recently, the selective state space model (SSM) has rivaled the previous backbone like vision transformer in performance while significantly reducing memory usage due to linear-scale complexity, showing robust generalization across 1D audio [24] and 2D image signals [42] when evaluated at varied frequencies. Given the inherent spatiotemporal richness due to events' high temporal resolution, a natural motivation arises for harnessing the exceptional power of SSM for event spatiotemporal modeling. To this end, we propose PASS, a novel framework for recognizing event streams capable of processing a broad distribution of event length ranging from $10^6$ to $10^9$ and generalizing across varying inference frequencies, as depicted in Fig. 1 . By harnessing the linear complexity and strong input frequency generalization of SSM, PASS delivers exceptional recognition performance and frequency generalization. It brings two key technical breakthroughs.

Firstly, since the large number of event length could cause difficulties for SSM in effectively learning the spatiotemporal properties from events, as SSM's hidden state updates rely heavily on the sequence length and feature order. To this end, we propose a novel Path-selective Event Aggregation and Scan (PEAS) module to aggregate and convert events into sequence features with fixed dimensions Concretely, as shown in Fig. 3, a selection mask is first learned from the original event frame representation to facilitate the frame selection. Then, the bidirectional event scan is conducted on the selected perimeters to convert them into sequence features. This adaptive process ensures the

event scan path is end-to-end learnable and responsive to every event input, thus enabling our PASS to effectively process event streams across a broad distribution of event length (Tab. 4).

Secondly, the varying sampling frequencies hinder the generalization of SSM during inference, as empirically verified in Tab. 5. This suggests that alterations in the input sequence length and order due to sampling frequency shifts greatly affect model performance. For this reason, we propose a novel Multi-faceted Selection Guiding (MSG) loss. It minimizes the randomness of the PEAS module event frame selection process caused by the random initialization of the selection mask's weight. Our MSG loss better facilitates alleviating the redundancy of the selected event frames, thus strengthening model generalization across varying inference frequencies (Tab. 5).

Extensive experiments across five public and three proposed datasets demonstrate PASS's superior performance. It outperforms previous methods by +3.45%, +0.38%, +8.31% +2.25% and +3.43% on the public PAF, SeAct, HARDVS, N-Caltech101, and N-Imagenet datasets, respectively. Additionally, PASS shows superior generalization across varying inference frequencies, with a maximum accuracy drop of -8.62% compared to -20.69% for previous methods. Given the absence of event-based recognition datasets with a large number of event length, we created two synthetic datasets and recorded one real-world dataset with around $10^9$ event length: ArDVS100 covers 100 action transitions with different meta-actions, TemArDVS100 features the same meta-actions yet in different combinations to evaluate the model's fine-grained temporal recognition ability, and Real-ArDVS10 dataset contains 10 recorded action transitions to assess the model's real-world generalization. Our PASS exhibits strong long spatiotemporal modeling across a broad distribution of event length ($1$-$10^9$), precise temporal perception, and effective generalization for real-world scenarios, achieving 97.35%, 89.00%, and 100% Top-1 accuracy on ArDVS100, TemArDVS, and Real-DVS10 datasets. Our main contribution can be summarized as follows:

- We propose PASS, a novel framework for recognizing events across a broad count distribution (event length range from $10^6$ to $10^9$) and generalizing to various inference frequencies.
- We introduce the PEAS module to convert asynchronous events into ordered sequence features, alongside MSG loss to promote effective event spatiotemporal modeling.
- Extensive experiments prove PASS's superior performance and strong inference frequency generalization. The proposed ArDVS100, TemArDVS100, and Real-ArDVS10 datasets prove the model's long spatiotemporal modeling, fine-grained temporal perception, and real-world effectiveness, respectively.

## 2  Related Works

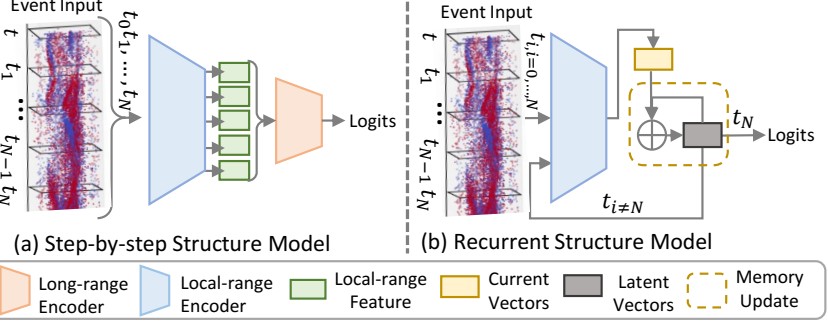

Figure 2: Comparison of two model structures for previous event-based recognition methods.

**Event-based Object / Action Recognition.** Existing event-based recognition works cover two main tasks: object recognition [75, 74, 15, 29, 73, 18, 25, 10, 35, 38] and action recognition [77, 66, 54, 65, 17, 48, 64, 39]. Specifically, object recognition captures stationary objects around $10^6$ events, whereas action recognition records dynamic human actions with approximate $e^7$ events. These methods tackle high temporal resolution event spatiotemporal complexity via two key approaches, as shown in Fig. 2: 1) step-by-step structure models and 2) recurrent structure models. Initially, the events are sampled into slices at fixed time intervals. The step-by-step structure models then use off-the-shelf backbones to extract local-range spatiotemporal features from event slices and then perform

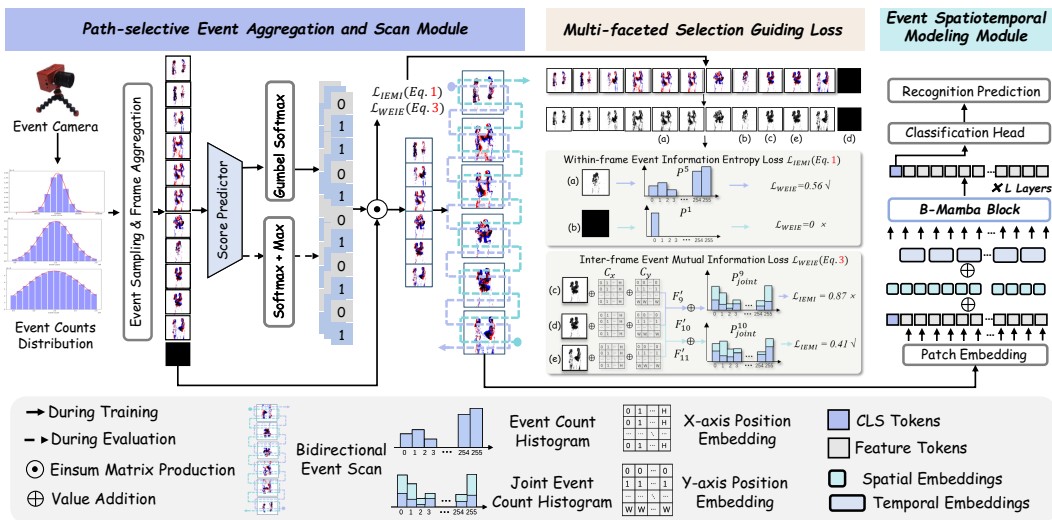

Figure 3: Overview of our proposed PASS framework.

long-range temporal modeling using various methods, such as simple average operation [77, 75], proposed modules [66, 69], and loss guidance [74, 30]. Recurrent structure models [54, 79], on the other hand, process the event slices sequentially, updating their hidden state based on the input at each time step. Both structures ensure adaptability to varying event lengths. However, step-by-step structure models struggle with high computational complexity, especially for handling more events in high-speed and long-duration scenarios. Recurrent structure models tend to forget the initial information due to their simplistic recurrent design and require longer training time due to their inability to process data in parallel. Additionally, as evidenced in Tab. 5, existing methods struggle to generalize across different inference frequencies, which is essential for applications in high-speed, dynamic visual scenarios [80]. In this work, we aim to improve event-based recognition across a broad distribution of event length with improved generalization across varying inference frequencies.

**State Space Model (SSM).** It has recently demonstrated considerable effectiveness in capturing the dynamics and dependencies of long sequences. Unlike transformers [3, 40] with quadratic complexity, SSMs [23, 59, 60, 14] offer superior performance through linear complexity and show robust generalization across 1D audio [24] and 2D image signals [42] when evaluated at varied frequencies. Mamba [22] distinguishes itself by introducing a data-dependent SSM layer, a selection mechanism, and hardware-level performance optimization. It motivates subsequent works in the vision [78, 67, 46], video [34, 45], and point cloud [72, 36] domains. Nikola *et al.* [80] first integrates SSMs with a recurrent ViT framework for event-based object detection to enhance the adaptability for varying sampling frequencies by low-pass band-limiting loss. Subsequent research explored applying SSMs, particularly Mamba [22], to event-based tasks, including action recognition [52, 5], tracking [26, 52, 62], detection [68], Unlike prior work, our work seeks to recognize event streams of broader distribution of event length and generalize across varying inference frequencies.

## 3 Preliminaries

**Event Stream.** Event cameras capture object movement by recording the pixel-level log intensity changes, rather than capturing full-frame at fixed intervals for conventional cameras. The asynchronous events, denoted as $\mathcal{E} = \{e_i(x_i, y_i, t_i, p_i)\}, i = 1, 2, ..., N$, reflects the brightness change $e_i$ for a pixel at the timestamp $t_i$, with coordinates $(x_i, y_i)$, and polarity $p_i \in \{1, -1\}$ [15, 73]. Here, 1 and -1 represent the positive and negative brightness changes.

**SSM for Vision.** SSMs [23, 59, 14, 60] originate from the principles of continuous systems that map an input 1D sequence $x(t) \in \mathbb{R}^L$ into the output sequence $y(t) \in \mathbb{R}^L$ through an underlying hidden state $h(t) \in \mathbb{R}^N$. Specifically, it is formalized by $dh(t)/dt = Ah(t) + Bx(t)$ and $y(t) =$

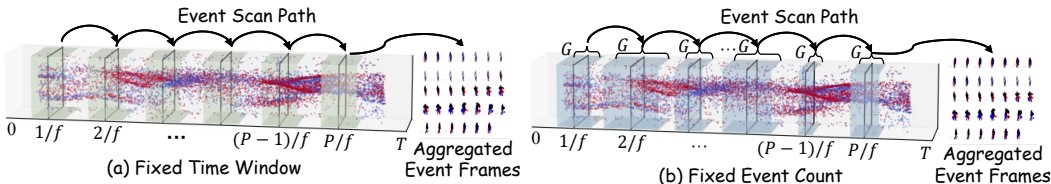

Figure 4: Illustration of event frame aggregation.

$Ch(t) + Dx(t)$, where $A \in \mathbb{R}^{N \times N}$, $B \in \mathbb{R}^{N \times 1}$, $C \in \mathbb{R}^{N \times 1}$, $D \in \mathbb{R}^{N \times 1}$ are the state matrix, the input projection matrix, the output projection matrix, and the feed-forward matrix.

## 4 Proposed Method

**Overview.** The PASS framework, as depicted in Fig. 3, processes events across a wide distribution of event length using our PEAS module and MSG loss, followed by the spatiotemporal modeling module for prediction. It comprises three components: (1) the PEAS module (Sec.4.1) for event sampling, event frame aggregation, and path-selective event selection. Then bidirectional event scan to encode events into sequence features with fixed dimensions. (2) On top of PEAS, the MSG loss $\mathcal{L}_{MSG}$ (Sec.4.2) is proposed for minimizing the randomness and redundancy of encoded features; (3) the event spatiotemporal modeling module (Sec.4.3) to predict the final recognition results.

### 4.1 Path-selective Event Aggregation and Scan (PEAS) Module

We aim to recognize event streams across a wide distribution of event length. The events are first converted into event presentations, where we select the event frame presentation with a fixed event length based on experiment results (see Sec. 5.3). The number of resulting aggregated event frames $P$ can vary greatly due to the high temporal resolution of events. This variability introduces complexity for spatiotemporal event modeling. Furthermore, due to SSM's recurrent nature, its hidden state update is greatly affected by the input sequence length and feature order, especially when modeling the long-range temporal dependencies. To reduce this variability, we propose our PEAS module, which consists of the following four components to encode events across a wider distribution of event length into sequence features with fixed dimensions in an end-to-end learning manner.

**Event Sampling and Frame Aggregation.** Unlike sequential language with compact semantics, events $\mathcal{E} = \{e_i(x_i, y_i, t_i, p_i)\} \in \mathbb{R}^{N \times 4}$, $i = 1, 2, ..., N$ denotes the asynchronous intensity change at the pixel $(x_i, y_i)$ at time $t_i$ with polarity $p_i \in \{1, -1\}$. The complexity of spatiotemporal event data requires efficient processing of this high-dimensional data. Following previous methods [75, 80, 2, 54], we sample events with duration $T$ at every fixed temporal windows $1/f$, where $f$ denotes the sampling frequency, *e.g.* 50 ms time windows $1/f$ corresponding to sampling frequency $f = 20Hz$. We group a number of events $G$ at each sampling time, as shown in Fig. 4 (b). This sampling method is more effective and robust than grouping events within fixed time windows as illustrated in Fig. 4 (a), as evidenced in the following Sec 5.3. Therefore, we obtain $P = Tf$ event groups $\mathcal{E}' \in \mathbb{R}^{P \times G \times 4}$. Then, we utilize the event frame representation [75] to transform the event groups $\mathcal{E}'$ into a series of event frames $F \in \mathbb{R}^{P \times H \times W \times 3}$. This transformation enables the use of traditional computer vision methods designed for frame-based data.

**Path-selective Event Scan.** With the aggregated event frame input $F$, we then conduct our path-selective event scan to reduce the variability of events. Concretely, as shown in Fig. 3, with the aggregated event frames $F \in \mathbb{R}^{P \times H \times W \times 3}$ as input, we utilize a lightweight score predictor composed of two 3D convolutional layers, followed by an activation function to generate a selection mask $M \in \mathbb{R}^{K \times P}$, where $K$ represents the number of selected frames and $P$ represents the number of original frames. The elements of $M$ are either 0 or 1, with each marking the position of the selected event frame. Due to the non-differentiable nature of the max operation applied after the standard Softmax function to produce class probabilities, we employ the differentiable Gumbel Softmax [28] for backpropagation during training. The standard Softmax is used for inference to facilitate the training process. Next, we utilize the Einsum matrix-matrix multiplication between the selection mask

$M$ and the original event frames $F$ to obtain the final selected event frames $F^{'} \in \mathbb{R}^{K \times H \times W \times 3}$. The above process ensures that $F^{'}$ can be derived from the original event frame input $F$ in an end-to-end learning manner. Next, with the obtained selected event frames $F^{'} \in \mathbb{R}^{K \times H \times W \times 3}$, we convert $F^{'}$ into a 1D sequence using the bidirectional event scan, following the spatiotemporal scan proposed in [34]. As illustrated in Fig. 3, this scan elegantly follows the temporal and spatial order, sweeping from left to right and cascading from top to bottom. In this way, unlike scanning the original $P$ event frames, our PEAS module can adaptively skip multiple event slices and encode the events across a wide event distribution ($10^6$ to $10^9$) into encoded features with fixed dimensions.

## 4.2 Multi-faceted Selection Guiding (MSG) Loss

While the proposed PEAS module is differentiable and capable of learning through back-propagation, the basic multi-class cross-entropy loss, $L_{CLS}$, is inadequate for effectively guiding model optimization. This is because the selection of event frames is stochastic at the onset of training due to the random weight initialization of the PEAS module. Consequently, during training, the model can only optimize performance based on the distribution of these randomly selected frames, rather than improving the PEAS module for adaptive selection of input events. To facilitate effective optimization, we propose the MSG loss that addresses two crucial challenges: 1) reducing the randomness of the selection process to ensure the selected sequence features can encapsulate the entirety of the sequence; and 2) guaranteeing that each selected event feature stands out from the others, thus eliminating redundancy. To be specific, the MSG loss comprises two components, which will be detailed in the subsequent subsections.

**Within-Frame Event Information Entropy (WEIE) Loss:** We introduce within-frame event information entropy loss $\mathcal{L}_{WEIE}$ to reduce the randomness of frame selection, which arises from the random initialization of the PEAS module (Sec. 4.1). $\mathcal{L}_{WEIE}$ quantifies the image information entropy of each event frame. As shown in Fig. 3, the WEIE loss for the padding frame Fig. 3 (b) is zero. In contrast, the WEIE loss for the non-padding frame Fig. 3 (a) is greater than zero. Intuitively, a higher WEIE loss indicates that the selected event frame contains more information and richer details. Thus, maximizing $\mathcal{L}_{WEIE}$ helps enhance model optimization to minimize randomness in the selection process. Specifically, we first calculate the frequency histogram $P^k = hist(gray(F_k^{'}))$ for each selected event frame $F_k^{'}$, where $K$ indicates the number of selected event frames, $gray(.)$ converts RGB event frames to grayscale and $hist(.)$ indicates histogram statistics frequency. Then the $\mathcal{L}_{WEIE}$ is calculated as follows:

$$\mathcal{L}_{WEIE} = -\sum_{k=1}^{K}\sum_{i=1}^{N} P_i^k \log P_i^k / K \qquad (1)$$

where $N$ is the number of histogram bins; $K$ indicates the number of selected event frames.

**Inter-frame Event Mutual Information (IEMI) Loss:** On top of the WEIE loss to quantify the information entropy for each event frame, we additionally propose the inter-frame event mutual information loss $\mathcal{L}_{IEMI}$ to reduce the redundancy among selected event frames. $\mathcal{L}_{IEMI}$ quantifies the mutual information [53] between every two consecutive event frames. As shown in Fig. 3, the $\mathcal{L}_{WEIE}$ for Fig. 3 (c) and Fig. 3 (d) are greater than the $\mathcal{L}_{WEIE}$ for Fig. 3 (d) and Fig. 3 (e). Intuitively, a lower IEMI loss indicates greater differences between the frames. Thus, minimizing IEMI loss guides the model in maximizing the difference between selected event frames. Specifically, $\mathcal{L}_{WEIE}$ is composed of the joint event length histogram $hist(.)$ between every two consecutive event frames $F_k^{'}$ and $F_{k+1}^{'}$, along with their spatial coordinates $C_x$ and $C_y$ to indicates the position information. We compute $\mathcal{L}_{WEIE}$ within every consecutive event frame $F^{'} \in \mathbb{R}^{K \times H \times W \times 3}$ to lower computational cost. The IEMI loss $\mathcal{L}_{IEMI}$ is formulated as follows:

$$P_{joint}^k = hist(gray(F_k^{'}) + gray(F_{k+1}^{'}) + C_x + C_y), \qquad (2)$$

$$\mathcal{L}_{IEMI} = -\frac{1}{K-1}\sum_{k=1}^{K-1}(\sum_{i=1}^{N}\sum_{j=1}^{N} P_{joint}^k(i,j) \times \log(P(i)P(j)/P_{joint}^k(i,j))), \qquad (3)$$

where $N$ indicates the number of histogram bins and $K$ is the number of selected event frames.

**Total Objective:** Given the final prediction class $y$ and the ground-truth class $Y$, the total objective is composed by the MSG loss $\mathcal{L}_{MSG}$ with three components and the commonly used multiclass cross-entropy loss $\mathcal{L}_{CLS}$:

$$\mathcal{L}_{total} = \underbrace{\mathcal{L}_{IEMI} - \mathcal{L}_{WEIE} +}_{\mathcal{L}_{MSG}} + \mathcal{L}_{CLS}(y, Y). \qquad (4)$$

### 4.3 Event Spatiotemporal Modeling Module

After the PEAS module followed by the MSG loss, event inputs are transformed into the event frame sequence $F^{'} \in \mathbb{R}^{K \times H \times W \times 3}$. Given the inherently longer sequences because of the event stream's high temporal resolution, we leverage the SSM for event spatiotemporal modeling with linear complexity. As shown in Fig. 3, we first employ the 3D convolution with kernel size $1 \times 16 \times 16$ for patch embedding to transform the event frames into $L$ non-overlapping spatiotemporal tokens $x_e \in \mathbb{R}^{L \times C}$, where $L = T_s \times H \times W / 16 \times 16$ and $C$ refer to feature dimensions. The SSM model, designed for sequential data, is sensitive to token positions, making preserving spatiotemporal position information crucial. Thus, we concatenate a learnable classification token $X_{cls} \in \mathbb{R}^{1 \times C}$ at the start of the sequence and then add a learnable spatial position embedding $P_s \in \mathbb{R}^{(1+L) \times C}$ and temporal embedding $P_t \in \mathbb{R}^{T_s \times C}$ to obtain the final input sequence $x = [x_{cls}, x_e] + P_s + P_t$. Next, the input sequence $x$ passes into $L$ layers of stacked B-Mamba blocks. [22]. Note that the bidirectional event scan is actually conducted in the B-Mamba blocks for code implementation. Finally, the [CLS] token is extracted from the final layer's output and forwarded to the classification head, which consists of the normalization layer and the linear classification layer for the final prediction $y$.

## 5 Experiments and Evaluation

### 5.1 Experiments settings

**Public Datasets:** Five publicly available event datasets are evaluated in this paper, including PAF [41], SeAct [77], HARDVS [63], N-ImageNet [29] and N-Caltech101 [43].

**Our ArDVS100, Real-ArDVS10 and TemArDVS100 Dataset**. Existing datasets only provide events within a limited distribution of event length ($10^6$ for objects and $10^7$ for actions). We introduce the ArDVS100 and TemArDVS across a broad distribution of event length ($10^6$ to $10^9$), synthesized by concatenating event streams with varying meta actions, thus **capturing action transitions over time**. Specifically, ArDVS100 and TemArDVS datasets contain 100 action classes, with event durations of 1$s$ to 256$s$ and 14$s$ to 214$s$ respectively. TemArDVS offers fine-grained temporal labels for more accurate action temporal recognition, distinguishing actions like 'sit down then get up' from 'get up then sit down,' while the ArDVS100 dataset treats them as the same. We allocated 80% for training and 20% for testing (evaluating). Additionally, to assess the model's real-world applicability, we created a real-world dataset, named Real-ArDVS10, comprising event-based actions lasting from 2$s$ to 75$s$, encompassing 10 distinct classes selected from the ArDVS100 datasets. The train and validation (test) split ratio is 7:3.

Table 1: Model structure settings.

| Model | Layer | Dim D | Param. | FLOPS(G) | Inference Time(ms) | FPS |
|---|---|---|---|---|---|---|
| Tiny (T) | 24 | 192 | 7M | 1.1 | 4.1 | 243.9 |
| Small (S) | 24 | 384 | 25M | 4.3 | 15.7 | 63.7 |
| Middle (M) | 32 | 576 | 74M | 12.7 | 40.4 | 24.7 |

**Model Architecture & Experimental Settings:** In alignment with ViT [13], we modify the depth and embedding dimensions to match models of comparable sizes, including Tiny (T), Small (S), and Middle (M). We adopt the pre-trained VideoMamba [34] model checkpoints for initialization. All ablation studies, unless specifically stated, use the Tiny version on the PAF dataset at a sampling frequency of 0.8 Hz with 16 selected event frames. We reproduced [80] from their official GitHub repository and evaluated it on our proposed and event-based recognition datasets for comparative analysis. The detailed model structure settings, parameter estimation, and computational complexity

Table 2: Comparison with previous methods for event-based object recognition.

| Object Recognition (Around $10^6$ events) | | | |
|---|---|---|---|
| Model | Param. | Top-1 Accuracy(%) | |
| | | N-Caltech101 | N-Imagenet |
| EST[19] | | 81.70 | 48.93 |
| EDGCN [9] | 0.77M | 83.50 | - |
| Matrix-LSTM [4] | - | 84.31 | 32.21 |
| E2VID [50] | 10M | 86.60 | - |
| DiST[29] | - | 86.81 | 48.43 |
| MEM [31] | - | 90.10 | 57.89 |
| S5-ViT-B-K(1) [80] | 17.5M | 88.32 | - |
| S5-ViT-B-K(2) [80] | 17.5M | 88.44 | - |
| EventDance [74] | 26M | 92.35 | - |
| PASS-T-$K$(1) | 7M | 88.29 | 48.74 |
| PASS-T-$K$(2) | | 89.72 | 48.60 |
| PASS-S-$K$(1) | 25M | 90.92 | 53.74 |
| PASS-S-$K$(2) | | 91.96 | 56.10 |
| PASS-M-$K$(1) | 74M | 94.20 | 61.12 |
| PASS-M-$K$(2) | | 94.60+2.25 | 61.32+3.43 |

Table 3: Comparison with previous methods for event-based action recognition.

| Action Recognition (Around $10^7$ events) | | | | |
|---|---|---|---|---|
| Model | Param. | Top-1 Accuracy(%) | | |
| | | PAF | SeAct | HARDVS |
| EV-ACT [17] | 21.3M | 92.60 | - | - |
| EventTransAct [7] | - | - | 57.81 | - |
| EvT [54] | 0.48M | - | 61.30 | - |
| TTPIONT [51] | 0.33M | 92.70 | - | - |
| Speck [71] | - | - | - | 46.70 |
| ASA [70] | - | - | - | 47.10 |
| ESTF [63] | - | - | - | 51.22 |
| S5-ViT-B-K(8) [80] | 17.5M | 92.93 | 58.21 | 74.85 |
| S5-ViT-B-K(16) [80] | 17.5M | 92.12 | 57.37 | 95.98 |
| ExACT [77] | 471M | 94.83 | 66.07 | 90.10 |
| PASS-T-$K$(8) | 7M | 91.38 | 51.72 | 98.40 |
| PASS-T-$K$(16) | | 94.83 | 49.14 | 98.37 |
| PASS-S-$K$(8) | 25M | 93.33 | 60.34 | 98.20 |
| PASS-S-$K$(16) | | 96.55 | 62.07 | 98.41+8.31 |
| PASS-M-$K$(8) | 74M | 98.28+3.45 | 65.52 | 98.05 |
| PASS-M-$K$(16) | | 96.55 | 66.38+0.38 | 98.20 |

Table 4: Results of event-based action recognition with around $10^6$ events).

| Arbitrary-duration Event Recognition (Around $10^9$ events) | | | | |
|---|---|---|---|---|
| Model | Param. | Top-1 Accuracy(%) | | |
| | | ArDVS100 | Real-ArDVS10 | TemArDVS100 |
| S5-ViT-B-K(16) [80] | 17.5M | 91.58 | 90.00 | 60.26 |
| S5-ViT-B-K(32) [80] | | 93.39 | 93.33 | 79.62 |
| PASS-T-$K$(16) | 7M | 90.20 | 80.00 | 59.20 |
| PASS-T-$K$(32) | | 93.85 | 93.33 | 89.00 |
| PASS-S-$K$(16) | 25M | 94.90 | 90.00 | 62.90 |
| PASS-S-$K$(32) | | 96.00 | 100.00 | 73.41 |
| PASS-M-$K$(16) | 74M | 96.00 | 93.33 | 71.06 |
| PASS-M-$K$(32) | | 97.35 | 100.00 | 82.50 |

are outlined in Tab. 1. Note that model parameters are estimates, changing with category count and selected event frames $K$.

## 5.2 Experiments Results

### 5.2.1 Event-based Recognition Results

**Results for recognizing event streams around $10^6$ events.** We evaluate PASS on N-Caltech101 and N-Imagenet. '$K$' indicates the number of selected event frames. As shown in Tab. 2, our PASS-M-K(2) secures a notable advantage, outperforming EventDance [74] by +2.25% for the N-Caltech101 dataset and MEM [31] by +3.43% for the N-Imagenet dataset. It achieves superior accuracy (+1.28%) with 2.5× fewer parameters (7M vs 17.5M) than S5-ViT-B-K(2) on the N-Caltech101 datasets, proving the superiority of PASS in effectively recognizing second-level event streams.

**Results for recognizing event streams around $10^7$ events.** Tab. 3 presents recognition results on three datasets with 1$s$ to 10$s$ event streams. Our PASS-M-K(2) outperforms previous methods, exceeding ExAct [77] by +3.45% and +0.38% on the PAF and SeAct datasets, respectively. Additionally, the PASS-S-K(16) achieves a remarkable 98.41% Top-1 accuracy on HARDVS dataset, surpassing ExAct [77] by +8.31% while using 25M parameters with reduced computational demands.

**Results for recognizing event streams around $10^9$ events.** In Tab. 4, we evaluate PASS on our ArDVS100, TemArDVS100, and Real-ArDVS10 datasets. On the ArDVS100 dataset, our PASS-M-K(32) attains 97.35% accuracy, outperforming [80] by 3.96% and highlighting its potential for arbitrary-duration event stream recognition. On the challenging TemArDVS100 dataset, our PASS-T-K(32) achieves 89.00% accuracy, surpassing [80] by 9.38% and demonstrating superior spatiotemporal action transition recognition. PASS-S-K(32) achieved 100% accuracy on the Real-ArDVS10 dataset, showcasing its effectiveness for real-world event-based action recognition.

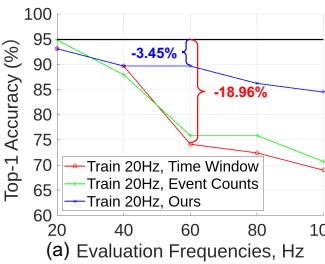 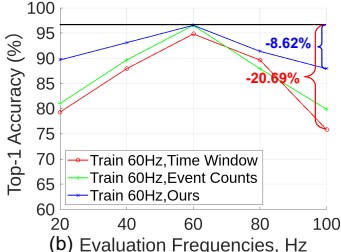 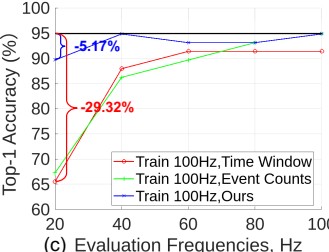

| (a) Evaluation Frequencies, Hz | (b) Evaluation Frequencies, Hz | (c) Evaluation Frequencies, Hz |

Figure 5: Model generalization results across varying inference frequencies $f$ training on PAF dataset with sampling frequencies at (a) 20Hz, (b) 60Hz, and (c) 100Hz.

Table 5: Ablation study on PEAS module & $\mathcal{L}_{MSG}$.

| Settings | PAF ($K(16)$) | ArDVS100 ($K(16)$) |
|---|---|---|
| | Top1(%) | Top1(%) |
| No Sampling | 92.90% | 92.31% |
| Random Sampling | 92.98% | 92.23% |
| PEAS | 93.33% | 92.84% |
| PEAS + $\mathcal{L}_{MSG}$ | **94.83%** | **93.85%** |

Table 6: Ablation study on $\mathcal{L}_{MSG}$.

| $\mathcal{L}_{MSG}$ | | | PAF($K(16)$) |
|---|---|---|---|
| $\mathcal{L}_{CLS}$ | $\mathcal{L}_{IEMI}$ | $\mathcal{L}_{WEIE}$ | Top1(%) |
| ✓ | ✗ | ✗ | 92.98% |
| ✓ | ✓ | ✗ | 93.75%+0.77 |
| ✓ | ✓ | ✓ | **94.83%**+1.85 |

### 5.2.2 Inference frequencies Generalization results.

**Datasets & Experimental settings** We trained PASS-S on the PAF dataset at 20 Hz, 60 Hz, and 100 Hz frequencies and evaluated across 20 Hz to 100 Hz to evaluate its inference frequency generalization. Two frame aggregation methods were considered as the baseline, namely fixed 'Time Windows' and fixed 'Event count'. (*Refer to Sec. 5.3 for more explanation and discussion.*)

**Results & Discussion** As shown in Fig. 5, regardless of whether the model is trained at low, medium, or high frequencies, our models demonstrate consistently strong performance across various inference frequencies, with a maximum performance drop of only 8.62% when our PASS model trained at 60 Hz and evaluated at 100 Hz. This finding underscores their robustness and generalizability compared to the baseline methods ('Time Windows' and 'Event count'), which experience significant performance declines, such as -18.96%, -20.69%, -29.32% for 'Time Windows' trained at 20 Hz, 60 Hz, and 100 Hz and evaluated at 60 Hz, 100 Hz, and 20 Hz, respectively.

### 5.3 Ablation Study

We perform ablation studies on our PASS framework to evaluate the effectiveness of the PEAS module (Sec. 4.1), $\mathcal{L}_{MSG}$ loss (Sec. 4.2).

**Impact of PEAS module & $\mathcal{L}_{MSG}$ loss.** As shown in Tab. 5, the baseline 'Random Sampling' randomly selects $K$ event frames and achieves 92.98% and 92.23% accuracy on the PAF and ArDVS100 datasets, respectively. By introducing PEAS, we improve accuracy to 93.33% and 92.84%, representing a performance gain of +0.35% and +0.61%, demonstrating its ability to preserve critical information. PEAS improves accuracy over baseline 'No Sampling' (+0.43% on PAF, +0.53% on ArDVS100), suggesting that the selected frames retain task-relevant information despite compression. When combining the PEAS module (Sec. 4.1) with $\mathcal{L}_{MSG}$ loss, the full model reaches 94.83% and 93.85% accuracy with a performance increase of +1.85% and +1.62% on PAF and ArDVS100 datasets, thus showing the effectiveness of $\mathcal{L}_{MSG}$ loss to reduce the randomness and redundancy of the encoded features.

**Effectiveness of Multi-faceted Selection Guiding Loss $\mathcal{L}_{MSG}$.** As presented in Tab. 6, we ablate the three components of $\mathcal{L}_{MSG}$ (Eq. 4). As the baseline, the $\mathcal{L}_{CLS}$ stands for the standard cross-entropy loss, which achieves a Top-1 accuracy of 92.98%. By employing the $\mathcal{L}_{IEMI}$ (Eq. 2), we attain 93.75% accuracy with 0.77% performance gain. The integration of $\mathcal{L}_{WEIE}$ (Eq. 1) yields an additional 1.85% increase in accuracy. In summary, all proposed components positively impact the final classification, thereby demonstrating their effectiveness.

Table 7: Ablation study on event representation.

| Representation | N-Caltech101 ($K(1)$) | | PAF ($K(16)$) | |
|---|---|---|---|---|
| | Top1(%) | Top5(%) | Top1(%) | Top5(%) |
| Frame(Gray) [75] | 90.48% | 97.53% | 93.33% | 100.00% |
| Frame(RGB) [75] | 90.94% | 97.82% | 94.83% | 100.00% |
| Voxel [11] | 90.19% | 97.02% | 92.47% | 100.00% |
| TBR [27] | 90.24% | 97.13% | 91.72% | 100.00% |
| EST [19] | 90.54% | 97.66% | 93.04% | 100.00% |

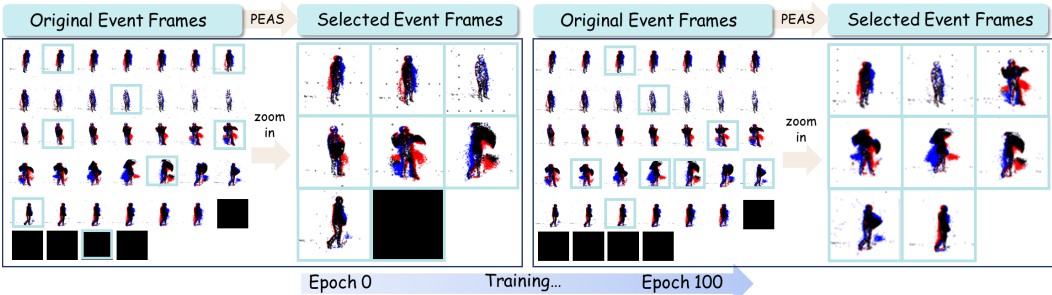

Figure 6: Visualization of PEAS module with MSG loss.

**Event representation.** Tab. 7 displays the impact of five commonly used event representations. The RGB frame [75] representation attains Top-1 accuracy rates of 90.94% on the N-Caltech101 dataset and 94.83% on the PAF dataset, surpassing the performance of the other three frame-based event representations, including gray frame [75], Voxel [11] and TBR [27] and one classical learnable event representation EST [19], validating the use of the RGB frame representation in the SSM model, as its pre-training image data has a smaller distribution gap with the RGB event frames.

**The visualization demonstration for the PEAS module.** Fig. 6 presents the original event frames and the $K$ selected ones at the start and end of the training process. The black parts indicate the padded zero-value frames among a batch. To accommodate varying event lengths and maintain consistent input sizes for batch training, frame padding is essential. In Fig. 6, the black parts represent the padded zero-valued frames within a mini-batch. At epoch 0, the PEAS module randomly selects event frames, resulting in unnecessarily padded frames and redundant event frames with repetitive information. After 100 epochs, the eight chosen frames exclude redundant frames and non-informative padding, demonstrating the effectiveness of the PEAS module and the MSG loss.

## 6   Conclusion

In this paper, we present our novel PASS framework for recognizing events. Extensive experiments prove that our PASS outperforms existing state-of-the-art approaches across five publicly available datasets. Our framework exhibits remarkable performance capabilities, successfully recognizing events across a wide event distribution ($10^6$ to $10^9$) as validated through our custom-developed ArDVS100, Real-ArDVS10, and TemArDVS datasets. Moreover, PASS also shows strong generalization across varying inference frequencies. We hope this method can pave the way for future model design for recognizing events for high-seed dynamic visual scenarios.

## Acknowledgements

This work is supported by the Science Foundation of China (NSF) under Grant No. 62206069 (affiliated with Guangzhou HKUST Fok Ying Tung Research Institute) and the MOE AcRF Tier 1 SSHR-TG Incubator Grant FY24 under Grant No. RSTG7/24.

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

# Appendix

In this appendix, we provide more details about model implementation, experimental settings, and datasets to complement the main paper. Additional analysis and discussions are also incorporated. Below is the table of contents:

# A  Model

## A.1  Technical Details of SSMs

State Space Models (SSMs) [23, 59, 14, 60] originate from the principles of continuous systems that map an input 1D sequence $x(t) \in \mathbb{R}^L$ into the output sequence $y(t) \in \mathbb{R}^L$ through an underlying hidden state $h(t) \in \mathbb{R}^N$. Specifically, it is formalized by $dh(t)/dt = Ah(t) + Bx(t)$ and $y(t) = Ch(t) + Dx(t)$, where $A \in \mathbb{R}^{N \times N}$, $B \in \mathbb{R}^{N \times 1}$, $C \in \mathbb{R}^{N \times 1}$, $D \in \mathbb{R}^{N \times 1}$ are the state matrix, the input projection matrix, the output projection matrix, and the feed-forward matrix.

$$dh(t)/dt = Ah(t) + Bx(t), \tag{5}$$

$$y(t) = Ch(t) + Dx(t), \tag{6}$$

where $A \in \mathbb{R}^{N \times N}$, $B \in \mathbb{R}^{N \times 1}$, $C \in \mathbb{R}^{N \times 1}$, $D \in \mathbb{R}^{N \times 1}$ are the state (or system) matrix, the input projection matrix, the output projection matrix and the feed-forward matrix.

The discretization process of SSMs is essential for integrating continuous-time models into deep-learning algorithms. [60]. We adopt Mamba [22] strategy, treating $D$ as fixed network parameters while introducing timescale parameter $\Delta$ to transform the continuous parameters $A$, $B$ into their discrete counterparts $\hat{A}$, $\hat{B}$, formulated as follows:

$$\hat{A} = exp(\Delta A) \tag{7}$$

$$\hat{B} = (\Delta A)^{-1}(exp(\Delta A) - I) \cdot \Delta B \tag{8}$$

$$h_t = \hat{A}h_{t-1} + \hat{B}x_t, \tag{9}$$

$$y_t = Ch_t. \tag{10}$$

Compared to previous linear time-invariant SSMs, Mamba proposed a selective scan mechanism that directly derived the parameters $B$, $C$, and $\Delta$ from the input during the training process, thus enabling better contextual sensitivity and adaptive weight modulation.

**Algorithm 1** PyTorch-style Pseudocode for the Proposed PEAS Module

```
# B, C, H, W: Batch size, Channel, Width, Height
# P, K: Amount of input and output event frames
# x: Input event frames with shape (B, P, C, H, W)
# y: Output selected frames with shape (B, K, C, H, W)

s = ScorePredictor(x) # Two-layer CNN network
# Predict scores for each event frame (B, K, P)
if self.training # Differentiable selection during training
    selection_mask = F.gumbel_softmax(pred_score, dim=2, hard=True)
else: # Hard selection during evaluation
    idx_argmax = s.max(dim=2, keepdim=True)[1]
    selection_mask = torch.zeros_like(s).scatter_(dim=2, index=idx_argmax, value=1.0)

B, K, P = selection_mask.shape
indices = torch.where(selection_mask.eq(1))
# Sort from largest to smallest corresponding to the time sequence
indices_sorted = torch.argsort(indices[2].reshape(B, K), dim=1)
# Rearrange mask based on temporal sequence
For i in range(B):
    selection_mask[i, :, :] = selection_mask[i, indices_sorted[i], :]

# Perform frame selection using the mask
y = torch.einsum('bkp, bcthw' → 'bcpkhw', selection_mask, x)
# Sum over time dimension
y = y.sum(dim=3) # (B,C,K,H,W)
```

## A.2 PyTorch-style Pseudocode for PEAS Module

In Algorithm 1, we present the PyTorch-style pseudocode of the proposed PEAS module to facilitate readers' understanding.

**Mask Selection (MS) Loss:** Due to the arbitrary length of event streams with different numbers of input event frames, frame padding is necessary to maintain consistent input sizes to ensure

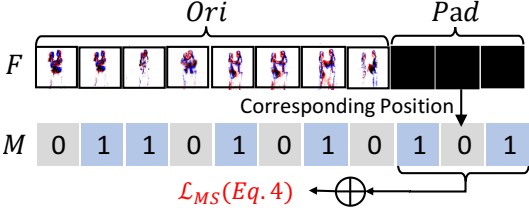

Figure 7: Illustration of components for the proposed MS loss.

training among a mini-batch. Therefore, we propose an MS loss $\mathcal{L}_{MS}$ to filter out the padded frames during selection. Specifically, as shown in Fig. 7, given the original event frame input $F \in \mathbb{R}^{P \times H \times W \times 3}$ and the selection mask $M \in \mathbb{R}^{K \times P}$, the $\mathcal{L}_{MS}$ loss sums the mask value $M_j, j = Ori + 1, ..., Ori + Pad$ at the corresponding position of the padding frame in $F_j, j = Ori+1, ..., Ori+Pad$, which is formulated as follows:

$$\mathcal{L}_{MS} = \sum_{i=1}^{K} \sum_{j=Ori+1}^{Pad} M_{i,j}/(K \times Pad),  \quad (11)$$

$K, Ori = P$, and $Pad$ indicate the number of selected event frames, original event frames, and padding frames, respectively.

# B  Experiments

## B.1  Experiment Settings

We utilize the default hyperparameters for the B-Mamba layer [78], setting the state dimension to 16 and the expansion ratio to 2. Additionally, we adjust the stochastic depth ratio to 0, 0.15, and 0.5 for the Tiny, Small, and Middle versions, respectively. We utilize the AdamW optimizer with a cosine learning rate schedule with the initial 5 epochs for linear warm-up. Unless a special statement is made, the default settings for the learning rate and weight decay are 1e-3 and 0.05, respectively. The model is trained with 100 epochs for PAF, SeAct, and N-Caltech101 datasets and 50 epochs for HARDVS, N-Imagenet, ArDVS100, TemArDVS100, and Real-ArDVS10 datasets. We employ BFloat16 precision during training to improve stability. For data augmentation, we implement random scaling, random cropping, random flipping, and data mixup of the event frames during the training phase.

## B.2  Event Frame Sampling Settings

The additional experiment settings of sampling frequency and aggregated event count per frame for different datasets are presented in Tab. 8.

Table 8: The sampling frequency and aggregated event count per frame for different datasets

| Dataset | Sampling Frequency | Aggregated Event Count / Frame |
|---|---|---|
| N-Caltech101 | 200 Hz | 50,000 |
| N-Imagenet | 50 Hz | 2,000,000 |
| PAF | 80 Hz | 100,000 |
| SeAct | 80 Hz | 80,000 |
| HARDVS | 100 Hz | 80,000 |
| ArDVS100 | 50 Hz | 80,000 |
| Real-ArDVS10 | 50 Hz | 80,000 |
| TemArDVS100 | 50 Hz | 80,000 |

## B.3  Reproduction Settings for [80]

We reproduced [80] from their official GitHub repository and evaluated it on our proposed and event-based recognition datasets for comparative analysis. The direct comparison is not feasible due to fundamental differences: **1)** task (object detection vs. object recognition), **2)** network structure (detection vs. classification head), **3)** SSM backbone (S4D, S5 vs. Mamba), **4)** evaluation datasets (detection vs. recognition), and **5)** evaluation metrics (mAP vs. accuracy).

For the above reasons, to ensure a fair comparison, we replaced our Mamba backbone with [80]'s S5-ViT-B model and substituted its YOLOX detection head with our classification head based on its GitHub repository PyTorch implementation. We did not use the S4D backbone due to its lower performance compared to the S5 model, as reported by [80]. We adopted [80]'s event voxel representation, creating event voxels based on 50 ms time windows corresponding to 20 Hz sampling frequency, divided into T = 10 discrete bins. Following [80], we applied data augmentation techniques such as random horizontal flips and zooming. The training was performed on the PAF and SeAct datasets for 100 epochs and on HARDVS ArDVS100, TemArDVS100, and Real-ArDVS10 datasets for 50 epochs. We also integrated the PEAS module, with $K$ indicating the number of selected event frames.

## B.4  More Experiment Results

**Frame Aggregation Method: Time Windows *vs.* Event count.** We illustrate two event aggregation methods in Fig. 4, where 'Event count' aggregation leads to varying aggregation temporal ranges and 'Time Windows' keeps them consistent. As shown in Fig. 5 (b), 'Event count' performs better compared to 'Time Windows'. For example, 'Event count' achieves **96.55%** accuracy compared to **94.83%** for 'Time Windows' when both trained and evaluated at 60 Hz. However, 'Event count' and 'Time Windows' experience **-16.66%** and **-18.97%** performance drops respectively, when evaluating

Table 9: Comparison of existing datasets with our ArDVS100 dataset.

| Dataset | Year | Sensors | Object | Scale | Class | Real | Temporal Fine-grained Labels | Duration(s) |
|---|---|---|---|---|---|---|---|---|
| MNISTDVS [43] | 2013 | DAVIS128 | Image | 30,000 | 10 | ✗ | ✗ | - |
| N-Caltech101 [43] | 2015 | ATIS | Image | 8,709 | 101 | ✗ | ✗ | 0.3s |
| N-MNIST [57] | 2015 | ATIS | Image | 70,000 | 10 | ✗ | ✗ | 0.3s |
| CIFAR10-DVS [33] | 2017 | DAVIS128 | Image | 10,000 | 10 | ✗ | ✗ | 1.2s |
| N-ImageNet [29] | 2021 | Samsung-Gen3 | Image | 1,781,167 | 1,000 | ✗ | ✗ | 0.1s |
| ES-lmageNet [37] | 2021 | - | Image | 1,306,916 | 1,000 | ✗ | ✗ | - |
| DvsGesture [1] | 2017 | DAVIS128 | Action | 1,342 | 11 | ✓ | ✗ | 6s |
| N-CARS [58] | 2018 | ATIS | Car | 24,029 | 2 | ✓ | ✗ | 0.1s |
| ASLAN-DVS [2] | 2019 | DAVIS240 | Hand | 100,800 | 24 | ✓ | ✗ | 0.1s |
| PAF [41] | 2019 | DAVIS346 | Action | 450 | 10 | ✓ | ✗ | 5s |
| HMDB-DVS [2] | 2019 | DAVIS240c | Action | 6,766 | 51 | ✗ | ✗ | 19s |
| UCF-DVS [2] | 2019 | DAVIS240c | Action | 13,320 | 101 | ✗ | ✗ | 25s |
| DailyAction [39] | 2021 | DAVIS346 | Action | 1,440 | 12 | ✓ | ✗ | 5s |
| HARDVS [63] | 2022 | DAVIS346 | Action | 107,646 | 300 | ✓ | ✗ | 5s |
| THUEACT50 [17] | 2023 | CeleX-V | Action | 10,500 | 50 | ✓ | ✗ | 2s-5s |
| THUEAC50CHL [17] | 2023 | DAVIS346 | Action | 2,330 | 50 | ✓ | ✗ | 2s-6s |
| Bullying10K [12] | 2023 | DAVIS346 | Action | 10,000 | 10 | ✓ | ✗ | 1s-20s |
| SeAct [77] | 2024 | DAVIS346 | Action | 580 | 58 | ✓ | ✗ | 2s-10s |
| DailyDVS-200 [61] | 2024 | DVXplorer Lite | Action | 22,046 | 200 | ✓ | ✗ | 2s-20s |
| ArDVS100 | 2024 | DAVIS346 | Action | 8,000 | 100 | ✗ | ✗ | 1s-263s |
| Real-ArDVS10 | 2024 | DAVIS346 | Action | 100 | 10 | ✓ | ✗ | 2s-75s |
| TemArDVS100 | 2024 | DAVIS346 | Action | 8,000 | 100 | ✗ | ✓ | 14s-214s |

Table 10: Model generalization results across different inference frequencies $f$ on PAF dataset.

| Train $f$ | Settings | Top-1 Accuracy & Performance Drop (%) | | | | |
|---|---|---|---|---|---|---|
| | | Val $f$ | | | | |
| | | 20 Hz | 40 Hz | 60 Hz | 80 Hz | 100 Hz |
| 20 Hz | Time Windows | 93.10 | $89.65_{-3.45}$ | $74.14_{-18.96}$ | $72.41_{-20.69}$ | $68.97_{-24.13}$ |
| | Event Counts | 94.83 | $87.93_{-6.90}$ | $75.86_{-18.97}$ | $75.86_{-18.97}$ | $70.69_{-24.14}$ |
| | Event Counts + PAST-SSM-S | 93.10 | $89.65_{-3.45}$ | $89.65_{-3.45}$ | $86.21_{-6.89}$ | $84.48_{-8.62}$ |
| 60 Hz | Time Windows | $79.31_{-15.52}$ | $87.93_{-6.90}$ | 94.83 | $89.65_{-5.18}$ | $75.86_{-18.97}$ |
| | Event Counts | $81.03_{-15.52}$ | $89.65_{-6.90}$ | 96.55 | $87.93_{-8.62}$ | $79.89_{-16.66}$ |
| | Event Counts + PAST-SSM-S | $89.66_{-6.89}$ | $93.1_{-3.45}$ | 96.55 | $91.38_{-5.17}$ | $87.93_{-8.62}$ |
| 100 Hz | Time Windows | $65.51_{-25.86}$ | $87.93_{-3.44}$ | $91.37_{-0}$ | $91.37_{-0}$ | 91.37 |
| | Event Counts | $67.24_{-27.59}$ | $86.21_{-8.62}$ | $89.65_{-5.18}$ | $93.1_{-1.73}$ | 94.83 |
| | Event Counts + PAST-SSM-S | $89.66_{-5.17}$ | $94,83_{-0}$ | $93.1_{-1.73}$ | $93.1_{-1.73}$ | 94.83 |

at 100 Hz. This result leads us to propose the PEAS module to improve model generalization across inference frequencies.

**The statistics result for model generalization across varying inference frequencies.** In Tab. 10, we present the specific statistics result for Fig.7 in the main paper for future comparison.

## C  Dataset Details

### C.1  ArDVS100 Dataset

ArDVS100 contains 100 different action series with varying durations synthesized by concatenating the randomly selected event streams from the HARDVS [63] dataset. The ArDVS100 contains 8000 event stream, which durations range from 1.46s to 263.26s, with a mean of 45.62s. To maintain brevity, Tab. 11 details only the selected 10 action series. We can observe that different classes feature distinct action series with varying meta-action counts, thus resulting in differing durations.

## C.2 Real-ArDVS10 Dataset

The Real-ArDVS10 dataset was recorded using the DVS346 event camera, which has a resolution of 346 × 240 pixels. The Real-ArDVS10 dataset includes ten action series randomly selected from the ArDVS100 dataset. During recording, participants stood before an event camera and performed meta-actions sequentially as instructed. Ten individuals (8 male, 2 female) contributed to the dataset, with detailed meta-action descriptions provided in Tab. 12.

## C.3 TemArDVS100 Dataset

ArDVS100 contains 100 different action series with varying durations synthesized by concatenating the randomly selected event streams from both HARDVS [63] and DailyDVS-200 [61] datasets. The ArDVS100 is made up of 8000 event streams, whose durations range from 14.53$s$ to 213.54$s$, with a mean of 93.87$s$. For presentation simplicity, we just illustrate the selected 8 action series with detailed action descriptions in Tab. 13. Classes 1 to 4 and 97 to 100 share the same four meta-actions but form distinct action series through varying meta-action combinations, allowing the TemArDVS100 dataset to provide fine-grained temporal labels for more precise action recognition.

## C.4 Dataset Comparision

We compare our proposed ArDVS100, Real-ArDVS10, and TemArDVS100 datasets with existing event-based recognition datasets. As shown in Tab. 9, previous datasets contain second-level event streams lasting from 0.1$s$ to 20$s$, while our proposed Real-ArDVS10 and TemArDVS100 datasets provide minute-level duration event streams lasting from 1$s$ to 265$s$, 2$s$ to 75$s$ and 14$s$ to 215$s$, respectively. We believe these proposed benchmarks will provide enhanced evaluation platforms for recognizing event streams of arbitrary durations and inspire further research in this field.

## C.5 Publicly Available Dataset

Five publicly available event-based datasets are evaluated in this paper as follows: **1) PAF** [41], also known as DVS Action, is an indoor dataset featuring 450 recordings across ten action categories lasting around 5$s$. **2) SeAct** [77] is a newly released dataset for event-based action recognition, covering 58 actions within four themes lasting around 2$s$-10$s$. This work uses only class-level labels despite available caption-level labels. **3) HARDVS** [63] is currently the largest dataset for event-based action recognition, comprising 107,646 recordings of 300 action categories. It also has an average duration of 5$s$ and a resolution of 346 × 260. **4) N-ImageNet** [29] is derived from the ImageNet-1K dataset, where the RGB images are displayed on a monitor and captured by a moving event camera. It includes 1,781,167 event streams with 480 × 640 resolution across 1,000 unique object classes. **5) N-Caltech101** [43] contains event streams captured by an event camera in front of a mobile 180 × 240 ATIS system [49] with the LCD monitor presenting the original RGB images in Caltech101. There are 8,246 samples comprising 300 ms in length, covering 101 different types of items.

# D Discussion

**Model Limitation.** We observe that larger VideoMamba tends to overfit during our experiments, resulting in suboptimal performance. This issue is not limited to our models but is also observed in VMamba [22] and VideoMamba [34]. Future research could explore training strategies such as self-distillation and advanced data augmentation to mitigate this overfitting.

**Broader Impact.** Recognition is a critical vision task with widespread applications like robot navigation. Traditional RGB-based methods can degrade due to motion blur and lighting variations. Event cameras exclusively capture moving objects, providing resilience to rapid motion and illumination changes while consuming minimal power. This method may be useful for high-level recognition tasks. As a data-driven approach, the method's performance is sensitive to data biases. Careful attention to the data collection process is essential to ensure reliable and accurate results.

Table 11: Meta-action descriptions for 10 selected action series classes in the ArDVS100 dataset. ArDVS100 includes 100 action series of varying durations, created by randomly concatenating event streams from HARDVS [63] to capture temporal action transitions.

| Class Index | Description | Class Index | Description |
|---|---|---|---|
| Class 1 | Action050- Step in Place ture | Class 10 | Action028- Bow Straight Arm Rowing true |
| Class 20 | Action177- Pinch waist
Action006- Upper and Lower Swing Arms
Action014- Alternate Front Kick | Class 30 | Action181- Shoulder Lift
Action273- Shoulder Wrap
Action215- Skew Head Biye |
| Class 40 | Action185- clench your fist and start running
Action195- Touch the back of the head
Action039- Forward and backward sliding steps
Action240- Standing Long Jump
Action206- Hip Up Kick Jump
Action199- Touch the forehead | Class 80 | Action111- Left iliopsoas muscle stretching
Action185- clench your fist and start running
Action043- Single Leg Jump
Action197- Touch the neck and tilt the head
Action199- Touch the forehead
Action120- Bare Hand Hard Pull Boat Action
Action026- Bow Side Flat Lift
Action032- Eavesdropping action
Action046- Press stapler
Action186- Rubbing Hands for Heating
Action237- Hard Pull Swing
Action243- Standing Right Rear Leg Lift
Action121- Wandering and Pacing
Action236- Eye Care Exercise
Action093- Leg bending side sitting
Action023- Side Lift
Action082- Shout
Action132- Strike Ten Step Fist |
| Class 50 | Action174- Chest Beating
Action021- Body flexion and rotation
Action013- 9th set of broadcast gymnastics kicking exercises
Action024- Side Lift Swivel Arm
Action097- Left Back Stretch
Action248- Standing Twist | | |
| Class 60 | Action072- Right hand raised
Action274- Chest Stretch
Action265- Arrow Squat Knee Lift
Action257- Simplified Tai Chi Tower Knee Depression Step
Action208- Knock Calculator
Action264- Arrow Squat Kick
Action170- Fist
Action119- Bare Hand Squat
Action069- Right Lunge Twist Stretch
Action024- Side Lift Swivel Arm
Action182- Knee Lift
Action052- Cross waist punch | Class 100 | Action083- Biting Lips true
Action025- Side Lunge Squat true
Action061- Right swag true
Action244- Standing Left Leg Lift true
Action115- Opening and closing steps true
Action046- Press stapler true
Action169- Wave true
Action099- Left Front Thigh Stretch true
Action247- Standing Jump Transformation true
Action119- Bare Hand Squat true
Action237- Hard Pull Swing true
Action111- Left iliopsoas muscle stretching true
Action144- Arm Press Down true
Action253- Simplified Tai Chi Twin Peaks Through Ears true
Action147- Scratch your ears and cheeks true
Action103- Left Bend true
Action289- Cover the Eyes and Lift the Legs true
Action106- Left hand circle true |
| Class 70 | Action022- Body Rotation Movement
Action133- Snap Fingers
Action050- Step in Place
Action018- Alternating Knee Strike
Action056- Holding cheeks with both hands
Action207- Salute
Action244- Standing Left Leg Lift
Action208- Knock Calculator
Action198- Touch waist and clip back
Action212- Comb Hair
Action067- Right thigh front stretch
Action008- 9th set of broadcast gymnastics side movements | | |

Table 12: Meta-action descriptions for action series classes in the Real-ArDVS10 dataset. This dataset, captured by an event camera, features real human action transitions for ten randomly selected action series from the ArDVS100 dataset classes.

| Class Index | Description | Class Index | Description |
|---|---|---|---|
| Class 4 | Action038- Front and rear foot pads | Class 72 | Action049- Jumping Rope in Place
Action250- Standing Touch Toe
Action127- Touch Shoulder
Action086- hiss action
Action116 - jumping jack
Action171- Cover Ears
Action277- selfie
Action023- Side Lift
Action168- Block the Sun
Action211- Mummy Jump
Action296- Applause
Action279- Take a step forward |
| Class 11 | Action169- Wave | | |
| Class 15 | Action149- Surrender | | |
| Class 27 | Action143- Twist waist
Action154- Wipe the Neck
Action292- Air Kiss | | |
| Class 30 | Action181- Shoulder Lift
Action273- Shoulder Wrap
Action215- Skew Head Biye | | |
| Class 36 | Action250- Standing Touch Toe
Action066- Right single swing arm
Action195- Touch the back of the head | | |
| Class 40 | Action185- clench your fist and start running
Action195- Touch the back of the head
Action039- Forward and backward sliding steps
Action240- Standing Long Jump
Action206- Hip Up Kick Jump
Action199- Touch the forehead | Class 90 | Action106- Left hand circle
Action174- Chest Beating
Action131- Tie Hair
Action082- Shout
Action240- Standing Long Jump
Action273- Shoulder Wrap
Action111- Left iliopsoas muscle stretching
Action212- Comb Hair
Action067- Right thigh front stretch
Action255- Simplified Tai Chi as if sealed off
Action047- In Place Wide Distance Run
Action116- jumping jack
Action244- Standing Left Leg Lift
Action031- Make a Face
Action108- Left Oblique Pull Down Half Squat
Action016- Alternate Front Kick Jump
Action078- Right iliopsoas muscle stretch |
| Class 56 | Action106- Left hand circle
Action199- Touch the forehead
Action013-9th set of broadcast gymnastics kicking exercises
Action250- Standing Touch Toe
Action105- Left hand raised
Action116 - jumping jack | | |

Table 13: Meta-action descriptions for 8 selected action series in the TemArDVS100 dataset. TemArDVS100 includes 100 action series of varying durations created by randomly combining event streams from HARDVS [63] and DailyDVS-200 [61]. TemArDVS100 features action series with identical meta-action but different combinations, enabling fine-grained temporal labeling of action transitions.

| Class Index | Description | Class Index | Description |
|---|---|---|---|
| Class 1 | Throw the ball in hand into the basket.
Turn off the tap of the water dispenser or sink.
Walk forward with your chest out and eyes looking straight ahead.
Action090- Head to Head Comparison true | Class 2 | Walk forward with your chest out and eyes looking straight ahead.
Turn off the tap of the water dispenser or sink.
Action090- Head to Head Comparison true
Throw the ball in hand into the basket. |
| Class 3 | Turn off the tap of the water dispenser or sink.
Throw the ball in hand into the basket.
Walk forward with your chest out and eyes looking straight ahead.
Action090- Head to Head Comparison true | Class 4 | Throw the ball in hand into the basket.
Action090- Head to Head Comparison true
Turn off the tap of the water dispenser or sink.
Walk forward with your chest out and eyes looking straight ahead. |
| | ...... | | |
| Class 97 | Action135- Playing Tai Chi true
Action272- Draw Circle at Elbow true
Action242- Standing Right Leg Lift true
Action211- Mummy Jump true
Tear a piece of paper.
Action063- Right humeral triple extension true
Raise one or both hands, make a fist, and extend it outward from the inside.
Put on the hat that is in hand or on the table. | Class 98 | Put on the hat that is in hand or on the table.
Action211- Mummy Jump true
Action135- Playing Tai Chi true
Raise one or both hands, make a fist, and extend it outward from the inside.
Action063- Right humeral triple extension true
Action272- Draw Circle at Elbow true
Tear a piece of paper.
Action242- Standing Right Leg Lift true |
| Class 99 | Action135- Playing Tai Chi true
Tear a piece of paper.
Action272- Draw Circle at Elbow true
Raise one or both hands, make a fist, and extend it outward from the inside.
Put on the hat that is in hand or on the table.
Action211- Mummy Jump true
Action063- Righthumeral triple extension true
Action242- Standing Right Leg Lift true | Class 100 | Tear a piece of paper.
Action063- Right humeral triple extension true
Action242- Standing Right Leg Lift true
Action135- Playing Tai Chi true
Action211- Mummy Jump true
Put on the hat that is in hand or on the table.
Action272- Draw Circle at Elbow true
Raise one or both hands, make a fist, and extend it outward from the inside. |

