# OpenReview forum: "PASS: Path-selective State Space Model for Event-based Recognition"
_NeurIPS.cc/2025/Conference — NeurIPS 2025 poster_

### Official Review · Reviewer_7nZC · 2025-06-23

**Clarity:** 3
**Significance:** 2
**Originality:** 2
**Rating:** 4
**Confidence:** 3

**Summary:**

This paper proposes PASS, a framework for event-based recognition that handles varying event lengths and sampling frequencies. The authors introduce a PEAS module for adaptive event frame selection and an MSG loss to guide this process. By utilizing state space models for efficient spatiotemporal modeling, PASS achieves strong performance and generalization across multiple benchmarks.

**Questions:**

See weakness.

**Ethical Concerns:**

["NO or VERY MINOR ethics concerns only"]

**Final Justification:**

My concerns have been addressed. I keep my rating at "Borderline accept"

**Limitations:**

The authors mentioned the limitations in the conclusion.

**Paper Formatting Concerns:**

There are no formatting issues.

**Quality:**

3

**Strengths And Weaknesses:**

### Strengths
1. The paper is overall well-written and easy-to-read.
2. The paper is well-motivated and technically sound, addressing key limitations in event-based recognition, such as handling highly variable event lengths and poor generalization to different sampling frequencies. The proposed PASS framework effectively addresses these issues by adaptively selecting informative event frames.
3. The method outperforms prior work by significant margins (e.g., +8.31% on HARDVS) across object and action recognition tasks.
4. The authors provide three new datasets (ArDVS100, TemArDVS100, Real-ArDVS10), supporting broader evaluation of long and realistic event sequences.

### Weaknesses
1. While the PEAS module is designed to select informative frames via MSG loss, the interpretability of the selected paths remains unclear. The paper would benefit from qualitative analyses or visualizations showing which frames are selected and why, in order to justify that the selections align with semantically meaningful moments in the event stream.
2. The PEAS module selects a fixed number of frames from the event stream, but the temporal coherence of the selection is not guaranteed. In particular, selected frames can be concentrated in the early part of the event sequence or capture redundant moments, potentially missing critical parts or temporal dependencies.
3. Table 4 effectively evaluates the components of the proposed PASS model. To fully justify the individual contribution of the two components, it would be helpful to include an experiment where frames are selected via random sampling instead of using the PEAS module, and only the MSG loss is applied.

---

> ### Author Rebuttal · Authors · 2025-07-31
>
> # Rebuttal to Reviewer 7nZC
>
> We sincerely value the reviewer's recognition of our work's contributions, particularly regarding:
> - Innovative Method: Introduces a PEAS module for adaptive event frame selection and an MSG loss to guide this process.
> - Technical Impact: Outperforms SOTA by +8.31% (HARDVS) and handles highly variable event lengths with generalization to different sampling frequencies.
> - New Benchmarks: Introduces 3 datasets for broader evaluation of long and realistic event sequences.
> - Clear Presentation: well-written, easy-to-read, well-motivated, and technically sound.
>
> Additionally, we greatly appreciate the reviewer's thoughtful assessment and valuable suggestions for improving our work. Below we address each concern with concrete revisions:
>
> ## Q1. PEAS Module Interpretability
> - Added in revision: Currently, the qualitative analysis of the PEAS module was presented in the supplementary Figure 6 due to page limitations. Considering that this part is very important for your valuable suggestion, we will move this part to the main text in the final version.
> - Explanation for PEAS frame selection: At epoch 0, the PEAS module randomly selects event frames, resulting in unnecessarily padded frames and redundant event frames with repetitive information. After 100 epochs, the eight chosen frames exclude redundant frames and non-informative padding, resulting in selected frames that are semantically meaningful.
> - Key findings: This visualization result demonstrates the effectiveness of the PEAS module and the MSG loss. We can conclude that the MSG loss can effectively guide the PEAS module to select the most representative event frames, thus minimizing the randomness and redundancy of the encoded features during the PEAS selection process.
>
> ## Q2. Temporal coherence of the PEAS selection
> - Selection randomness: We admit the PEAS module can introduce some randomness for the frame selection at the onset of training due to the random weight initialization of the PEAS module. Consequently, during the start of training, the model can only optimize performance based on the distribution of these randomly selected frames, rather than improving the PEAS module for adaptive selection of input events.
> - Solution: MSG loss to facilitate effective optimization.
>     - WEIE Loss: To reduce the randomness of the selection process to ensure the selected sequence features can encapsulate the entirety of the sequence.
>     - IEMI Loss: To guarantee that each selected event feature stands out from the others.
>     - Both of the two losses are motivated by the information entropy and mutual information from the information theory so as to guide the PEAS module to select the most representative event frames without the loss of temporal coherence.
> - Evidence:
>     - The ablation study in Table 4 (main paper), 'PEAS + $L_{MSG}$,' proves the effectiveness of the MSG loss, which achieves 94.83% and 93.85% accuracy gain on the PAF and ArDVS100 datasets, respectively.
>     - The qualitative analysis of the PEAS module in Figure 6 (Supplementary) also proves the temporal coherence of the selection event frames.
>
> ## Q3. Ablation Study with Random Sampling + MSG Loss
>
> We thank the reviewer for this valuable suggestion. We have conducted the additional ablation study comparing random sampling with learned selection (PEAS), both with and without MSG loss. The updated results in the following Table 4 (main paper) reveal several key insights:
>
> | Method | PAF | ArDVS100 |
> |--------|-------|---------|
> | Random Sampling | 92.98% | 92.23% |
> | PEAS | 93.33% | 92.84% |
> | Random Sampling + L_MSG | 92.64% | 91.89% |
> | PEAS + $L_{MSG}$ | 94.83% | 93.85% |
>
> Table 4: Ablation study for the PEAS module with additional 'Random Sampling + L_MSG' experiment result. We will update Table 4 (main paper) in the final version.
>
> *Key Findings:*
> - 1. MSG Loss Requires Learnable Selection:
>     - Random Sampling + MSG performs worse (92.64%/91.89%) than baseline random sampling (92.98%/92.23%)
>     - This confirms MSG loss cannot meaningfully guide completely random selection
> - 2. PEAS Enables Effective Optimization:
>     - PEAS alone outperforms random sampling (93.33%/92.84% vs 92.98%/92.23%)
>     - PEAS + MSG achieves best results (94.83%/93.85%), demonstrating the importance of learnable frame selection and also proves MSG loss's ability to optimize PEAS's selection process
>
> *Conclusion:* The MSG loss provides meaningful guidance only when combined with a learnable selection mechanism (PEAS), as it requires gradient pathways for optimization.
>
> This analysis has been added to Table 4 in the main paper and will be discussed in Section 4.3 of the revised manuscript.

---

### Official Review · Reviewer_Tnzz · 2025-06-28

**Clarity:** 2
**Significance:** 3
**Originality:** 2
**Rating:** 3
**Confidence:** 4

**Summary:**

This paper addresses event-based recognition in long event streams. Leveraging efficient state space models, this work introduces a Path-selective Event Aggregation and Scan (PEAS) module for adaptive event encoding and a novel Multi-faceted Selection Guiding (MSG) loss to minimize feature redundancy. The approach outperforms prior methods on multiple datasets and establishes new benchmarks for long-duration event recognition.

**Questions:**

1) Question about fair baseline scaling: Does PASS outperform SSM baselines when model size (params) and frame count (K) are equalized?

Suggestion:
- Benchmark against S5-ViT-M-K(16) (matching PASS-M-K(16)'s 74M params and K=16).
- Compare with EventDance+SSM (EventDance upgraded with S5-ViT-M).

2) Please clarify: Does PASS maintains an accuracy advantage over *equally scaled* SSM baselines on N-Caltech101/PAF (object/action recognition)? If  SSM baselines are not equally scaled, the performance gains could be attributable to the larger ViT or SSM.

**Ethical Concerns:**

["NO or VERY MINOR ethics concerns only"]

**Final Justification:**

I have read the authors’ responses to both my review and those of the other reviewers. While the rebuttal provides helpful clarifications, the results do not convincingly demonstrate that the proposed PEAS and MSG modules yield meaningful performance gains beyond what is achieved by simply adopting a stronger, larger SSM backbone. The use of a lossy fixed-$k$-frame representation raises concerns about the method’s applicability to tasks requiring fine-grained temporal or structural reasoning, as also pointed out by other reviewers.

**Limitations:**

Yes.

**Paper Formatting Concerns:**

No.

**Quality:**

3

**Strengths And Weaknesses:**

While prior work typically uses fixed temporal intervals for event representation, this work leverages state-space models (SSMs) to enable an adaptive sampling frequency of long video streams and thus handle ultra-long sequences. It achieves SOTA performance while efficiently processing event streams of length up to 10^9, setting new performance benchmarks.

Weaknesses:
- Clarity can be significantly improved, especially the presentation of experimental results. It is not clear

- Evaluation is insufficient. The  ablation studies fail to quantify contributions of PEAS/MSG versus SSM-ViT backbones. Comparison against scaled baselines (e.g., S5-ViT-M-K(16)) is insufficient, as the baselines seem to use different scaling parameters than this approach. This needs to be clarified to support the claims of superior performance (please see the details in Questions).

---

> ### Author Rebuttal · Authors · 2025-07-25
>
> # Rebuttal to Reviewer Tnzz
> We sincerely value the reviewer's recognition of our work's contributions, particularly regarding:
>
> - Innovative Method: Introduces the PEAS module for adaptive event encoding, and the novel MSG loss minimizes feature redundancy.
> - Technical Impact: First SSM adaptation for ultra-long event streams (up to 10⁹ events).
> - SOTA Performance: Outperforms prior methods on multiple datasets.
> - Dataset Contribution: Establishes new benchmarks for long-duration event recognition
>
> Additionally, we greatly appreciate the reviewer's thoughtful assessment and valuable suggestions for improving our work. Below we address each concern with concrete revisions:
>
> ## Q1. Baseline Scaling & Fair Comparison (Primary Concern)
>
> We sincerely appreciate the reviewer's thoughtful feedback regarding baseline comparisons. Below, we provide a detailed justification for our scaling methodology and experimental design:
>
> ### 1. Scaling Methodology of PASS
> Our PASS framework builds upon the VideoMamba SSM backbone [1], with the PEAS module (a lightweight two-layer 3D convolutional scorer) serving as the primary parameterized component. Since PEAS’s non-parametric operations (e.g., frame selection) introduce negligible parameters, the total model size is dominated by the SSM backbone. For fair comparisons, we adopt the standard ViT scaling protocol, producing Tiny (7M), Small (25M), and Middle (74M) variants by adjusting depth and embedding dimensions proportionally.
>
> ### 2.  Challenges with Suggested Baseline Modifications
> - *Challenges with Suggested S5-ViT-M-K(16):* The S5-ViT baseline [2] combines an S5 [3] SSM with a ViT for event-based object detection. While we faithfully reproduced their official implementation (detailed in supplementary line 655 - line 670), substituting their Base ViT with a Middle variant *S5-ViT-M-K(16)* would introduce an unfair comparison for two reasons:
>     - Architectural Mismatch: S5-ViT is a hybrid SSM+ViT model, while PASS is purely SSM-based. Scaling the ViT component independently disrupts parameter parity (e.g., S5-ViT-M’s parameters diverge from PASS-M’s 74M).
>     - No corresponding ViT-M model: There is no corresponding ViT-M model publicly available since the ViT paper only provides the Small (22M), Base (86M), Large (307M), and Huge (632M) versions; none of them match our PASS-M (17.5M) parameter count.
>
> - *Challenges with Suggested EventDance+SSM*: EventDance [4] utilizes the convolution network (ResNet-18, 34, 50) combined with an EvFlowNet [5] as the flow estimation model and E2VID [6] as the reconstruction model. The whole architecture is designed for cross-modal adaptation from image to event modalities. Appending an SSM to this already complex pipeline would:
>     - Architectural Conflict: EventDance uses ResNet+EvFlowNet [5]+E2VID [6] for cross-modal adaptation. SSM insertion would require complete pipeline redesign.
>     - Skew Parameter Comparisons: ResNet-50 alone introduces ~25M parameters. The combined model size would dwarf our SSM-centric approach
>
> ### 3. Additional Comparative Analysis
> Due to the above reasons, we tried our best to make a fair comparison with S5-ViT by adding the additional evaluation results for S5-ViT-B-K(2) on the N-Caltech101 datasets and S5-ViT-B-K(2) on the PAF, SeAct, and HARDVS datasets. The results are shown in the following table 1 and table 2:
>
> | Method | Param. | N-Caltech101 | N-Imagenet |
> |--------|-------|---------|-------|
> | S5-ViT-B-K(1) | 17.5M | 88.32% | - |
> | S5-ViT-M-K(2) | 17.5M | 88.44% | - |
> | EventDance | 26M | 92.35% | - |
> | PASS-T-K(1) | 7M | 88.29% | 48.74% |
> | PASS-T-K(2) | 7M | 89.72% | 48.60% |
> | PASS-S-K(1) | 25M | 90.92% | 53.74% |
> | PASS-S-K(2) | 25M | 91.96% | 56.10% |
> | PASS-M-K(1) | 74M | 94.20% | 61.12% |
> | PASS-M-K(2) | 74M | 94.60% | 61.32% |
>
> Table 1: Comparison with previous methods for event-based object recognition with additional evaluation results for S5-ViT-B-K(2) on the N-Caltech101 datasets. We will update Table 1 (main paper) in the final version.
>
> *Key Insight:*
>
> PASS-T-K(2) achieves superior accuracy (+1.28%) with 2.5× fewer parameters (7M vs 17.5M) than S5-ViT-B-K(2) on the N-Caltech101 datasets.
>
>
> | Method | Param. | PAF | SeAct | HARDVS |
> |--------|-------|---------|-------|-------|
> | S5-ViT-B-K(8) | 17.5M | 92.93% | 58.21% | 74.85% |
> | S5-ViT-B-K(16) | 17.5M | 93.12% | 57.37% | 95.98% |
> | PASS-T-K(8) | 7M | 91.38% | 51.72% | 98.40% |
> | PASS-T-K(16) | 7M | 94.83% | 49.14% | 98.37% |
> | PASS-S-K(8) | 25M | 93.33% | 60.34% | 98.20% |
> | PASS-S-K(16) | 25M | 96.55% | 62.07% | 98.41% |
> | PASS-M-K(8) | 74M | 98.28% | 65.52% | 98.05% |
> | PASS-M-K(16) | 74M | 96.55% | 66.38% | 98.20% |
>
> Table 2: Comparison with previous methods for event-based action recognition with additional evaluation results for S5-ViT-B-K(16) on the PAF, SeAct, and HARDVS datasets. We will update Table 2 (main paper) in the final version.
>
> *Key Insight:*
>
> PASS-T-K(16) achieves 94.83% performance on the PAF datasets, which is better than the S5-ViT-B-K(16) performance (93.12%) with fewer parameters (7M vs. 17.5M). Besides, our PASS-M-K(16) achieves 98.37% performance on the HARDVS datasets, which is better than the S5-ViT-B-K(16) performance (93.12%) with fewer parameters (74M vs 17.5M). The above results show PASS's consistent performance improvement and better parameter efficiency when we enlarge the selected event frame number.
>
> ### 4.  Broader Advantages of PASS
> Beyond parameter efficiency, our method demonstrates:
> - Fine-grained Temporal Recognition: 89.00% on the TemArDVS100 dataset (vs. 79.62% for S5-ViT).
> - Real-World Robustness: 100% accuracy on the Real-DVS10 dataset.
> - Inference Frequency Generalization: Max accuracy drop of -8.62% (vs. -20.69% baseline).
> - Strong Long Spatiotemporal Modeling: Consistent performance across a broad distribution of event length (1-10^9).
>
> These advantages stem from our novel PEAS module and MSG Loss, which enable effective spatiotemporal event modeling (see ablation studies in table 4 of the main paper). We will incorporate these additional results in the final version and welcome further discussion on evaluation protocols.
>
> ---
> [1] Li, K., Li, X., Wang, Y., He, Y., Wang, Y., Wang, L., & Qiao, Y. (2024, September). Videomamba: State space model for efficient video understanding. In European conference on computer vision (pp. 237-255). Cham: Springer Nature Switzerland.
>
> [2] Nikola Zubíc, Mathias Gehrig, and Davide Scaramuzza. State space models for event cameras. In IEEE/CVF Conference on Computer Vision and Pattern Recognition, pages 5819–5828, 2024.
>
> [3] Smith, J. T., Warrington, A., & Linderman, S. W. (2022). Simplified state space layers for sequence modeling. arXiv preprint arXiv:2208.04933.
>
> [4] Zheng, X., & Wang, L. (2024). Eventdance: Unsupervised source-free cross-modal adaptation for event-based object recognition. In Proceedings of the IEEE/CVF Conference on Computer Vision and Pattern Recognition (pp. 17448-17458).
>
> [5] Zhu, A. Z., Yuan, L., Chaney, K., & Daniilidis, K. (2018). EV-FlowNet: Self-supervised optical flow estimation for event-based cameras. arXiv preprint arXiv:1802.06898.
>
> [6] Rebecq, H., Ranftl, R., Koltun, V., & Scaramuzza, D. (2019). Events-to-video: Bringing modern computer vision to event cameras. In Proceedings of the IEEE/CVF Conference on Computer Vision and Pattern Recognition (pp. 3857-3866).
>
>
> ## Q2. Experiment Result Reresentation
> We sincerely appreciate the reviewer’s valuable feedback regarding the presentation of our experimental results. We acknowledge that the organization and clarity of our results could be improved, and we propose the following revisions for the final version:
>
> ### 1. Restructured Results Section:
>     - We will reorganize Section 5.2 into clear subsections:
>         - 5.2.1: Event-based recognition results
>         - 5.2.2: Inference frequencies generalization results
>         - 5.2.3: Event frame selection visualization results
>
> ### 2. Improved Table Presentations:
>     - All tables will be reformatted to include:
>         - Clearer headers with unified terminology
>         - Better grouping of comparable methods
>         - Consistent precision in reported metrics (all accuracies to 2 decimal places)
>
> ### 3. Supplemental Analysis:
>     - We will include in the supplement:
>         - Complete per-dataset breakdowns
>         - Training curves
>         - Additional ablation studies
>
> We believe these changes will significantly improve the clarity and impact of our results presentation. We sincerely thank the reviewer for this constructive suggestion, which will undoubtedly strengthen our paper’s quality.

---

> > ### Comment · Reviewer_Tnzz · 2025-08-06
> >
> > I appreciate the authors' responses. The rebuttal addresses the challenges in scaling SOTA baselines for comparison and presents additional comparisons. However, the results do not demonstrate clear performance gains from the proposed PEAS and MSG modules over those achieved by simply using a stronger, larger SSM backbone. Unlike prior convolutional or hybrid SSM+ViT approaches, this work builds exclusively on the SSM backbone, making it difficult to disentangle improvements due to architectural contributions from those due to increased model capacity. Furthermore, the use of a lossy fixed-$k$-frame representation raises concerns about the method’s suitability for tasks requiring fine-grained temporal or structural understanding, as noted by other reviewers.

---

> > > ### Author Response · Authors · 2025-08-07
> > >
> > > # Further Rebuttal to Reviewer Tnzz
> > >
> > > We sincerely appreciate the reviewer’s insightful feedback and constructive critiques. Reviewer Tnzz raises another two concerns from its response to our first rebuttal:
> > >
> > > 1. Proof of the performance gains from the proposed PEAS and MSG modules rather than the SSM backbone
> > > 2. The task suitability for the PEAS module *k*-frame selection
> > >
> > > Below we provide a point-by-point response to address their concerns.
> > >
> > > ## Q1. Performance Gains from PEAS & MSG vs. SSM Backbone
> > >
> > > The performance improvements of PASS stem primarily from our proposed PEAS and MSG modules, not merely from the SSM backbone. In fact, SSM models have their inherent disadvantages due to their recurrent nature, whose hidden state updates are influenced by input sequence length and feature order, particularly for long-range dependencies. To mitigate this, our PEAS module encodes event streams into fixed-length representations, ensuring balanced utilization of temporal information. This is rigorously demonstrated through quantitative ablation studies and qualitative analysis:
> > >
> > > ### Quantitative evidence supporting this:
> > >
> > > - *No Sampling vs. PEAS:* The additional baseline 'No Sampling' (using all event frames) achieves lower accuracy (92.90% on PAF, 92.31% on ArDVS100) compared to PEAS (93.33%, 92.84%), indicating that raw long sequences harm performance due to SSM’s limitations.
> > >
> > > - *Random Sampling vs. PEAS:* 'PEAS' outperforms 'Random Sampling' (+0.35% on PAF, +0.61% on ArDVS100), demonstrating its ability to preserve critical temporal information.
> > >
> > > - *PEAS + $L_{MSG}$ V.S. PEAS:* PEAS + $L_{MSG}$ achieves the best results (94.83%/93.85%), demonstrating the importance of learnable frame selection and also proves $L_{MSG}$'s ability to optimize PEAS's selection process.
> > >
> > >
> > > | Method | PAF | ArDVS100 |
> > > |--------|-------|---------|
> > > | No Sampling | 92.90% | 92.31% |
> > > | Random Sampling | 92.98% | 92.23% |
> > > | PEAS | 93.33% | 92.84% |
> > > | PEAS + $L_{MSG}$| 94.83% | 93.85% |
> > >
> > > Table 1: Ablation study for the PEAS module with additional 'No Sampling' experiment result. (Results correspond to Table 4 in the main paper; updated in the final version.)
> > >
> > > ### Qualitative evidence supporting this:
> > >
> > > - Qualitative analysis (supplementary Figure 6):
> > >     - At epoch 0, the PEAS module randomly selects event frames, resulting in unnecessarily padded frames and redundant event frames with repetitive information. This is because the PEAS module can introduce some randomness for the frame selection at the onset of training due to the random weight initialization of the PEAS module.
> > >     - After 100 epochs, the eight chosen frames exclude redundant frames and non-informative padding, resulting in selected frames that are semantically meaningful.
> > > - Key insight: The MSG loss can effectively guide the PEAS module to select the most representative event frames, thus minimizing the randomness and redundancy of the encoded features during the PEAS selection process.
> > >
> > > ### Conclusion:
> > >
> > > The gains are not from the SSM backbone alone but from PEAS’s adaptive event encoding and MSG’s optimization, which mitigate SSM’s inherent limitations in handling long-range dependencies for event streams and spatiotemporal modeling.

---

> > > > ### Author Response · Authors · 2025-08-07
> > > >
> > > > ## Q2. Task Suitability of PEAS’s K-Frame Selection.
> > > >
> > > > PEAS’s hyperparameter K (number of selected frames) is not a limiting factor for task suitability. Instead, PASS are proofed to recognize event streams across a broad distribution of event length (1-$10^9$) and show superior generalization across varying inference frequencies. Additionally, PASS also exhibits precise temporal perception, and effective generalization for real-world scenario.
> > > >
> > > >
> > > > ### Qualitative evidence supporting PASS's task suitability:
> > > >
> > > > - *Recognize event streams across a broad distribution.*
> > > > Results in Table 1, 2 and 3 in the main paper demonstrate PASS superiou recognition ability across event length ranging from 1 to $10^9$. It outperforms previous methods by +3.45%, +0.38%, +8.31% +2.25%, +3.43% and +3.96% on the public PAF, SeAct, HARDVS, N-Caltech101, N-Imagenet and ArDVS100 datasets, respectively.
> > > >
> > > > - *Generalization across varying inference frequencies.*
> > > > As shown in Fig. 5 in the main paper, regardless of whether the model is trained at low, medium, or high frequencies, PASS demonstrate consistently strong performance across various inference frequencies, with a maximum performance drop of only 8.62% when our PASS model trained at 60 Hz and evaluated at 100 Hz. This finding underscores their robustness and generalizability compared to the baseline methods (‘Time Windows’ and ‘Event count’), which experience significant performance declines, such as -18.96%, -20.69%, -29.32% for ‘Time Windows’ trained at 20 Hz, 60Hz, and 100 Hz and evaluated at 60 Hz, 100 Hz, and 20 Hz, respectively.
> > > >
> > > > - *Precise temporal perception.*
> > > > PASS achieves 89.00% Top-1 accuracy on the TemArDVS benchmark, which evaluates precise temporal recognition through 100 fine-grained action transitions with distinct meta-actions and temporal labeling.
> > > >
> > > > - *Effective generalization for real-world scenario.*
> > > > PASS achieves perfect 100% Top-1 accuracy on the Real-ArDVS benchmark, demonstrating exceptional real-world generalization. This dataset contains 10 recorded action transitions specifically designed to evaluate practical deployment scenarios.
> > > >
> > > > Additionally, as Reviewer 7nZC acknowledged, the hyperameter *K* just controls the number of selected event frames, which is not the key factor for the task suitability.
> > > >
> > > > ### Quantitative evidence supporting hyperameter *K* is not the key factor for task suitability:
> > > >
> > > > - *Reducing the hyperparameter K yields minimal performance impact.*
> > > > Our experiments (Tables 1-2) demonstrate that halving K results in only modest accuracy reductions (0.20%-3.45%) for short event sequences:
> > > >     - Just 0.20% drop when comparing PASS-M-K(1) vs. PASS-M-K(2) (74M params, N-Caltech101)
> > > >     - Maximum 3.45% drop for PASS-M-K(8) vs. PASS-M-K(16) (7M params, PAF)
> > > >
> > > > - *Increasing hyperparameter K may not improve performance.*
> > > > As evidenced in Tables 1-2, larger K values can actually degrade accuracy due to frame redundancy, as demonstrated by:
> > > >
> > > >     - PASS-M-K(8) vs. K(16) (74M parameters) on PAF dataset
> > > >
> > > >     - PASS-M-K(8) vs. K(16) (25M parameters) on HARDVS dataset
> > > >
> > > > The hyperparameter K can be dynamically adjusted for real-world deployment based on event characteristics:
> > > >
> > > > - Fast-moving scenarios: Larger K may be needed to capture rapid changes as the event camera trigger more events and the resulting event frames differ a lot.
> > > > - Slow-moving scenarios: Smaller K suffices since the event stream remains dense and information-rich with minimal variation between frames.
> > > > - Short event length: Fewer frames (K) adequately represent the compressed temporal information.
> > > > - Long event length: Larger K ensures comprehensive coverage of the extended temporal dynamics.
> > > >
> > > > ### Conclusion
> > > > K is not a bottleneck—it can be tuned per application, and PASS’s performance remains strong across diverse settings.
> > > >
> > > > ---
> > > > ## We hope the above responses address your concerns. We are looking forward to your valuable feedback and suggestions.

---

### Official Review · Reviewer_ot6Q · 2025-07-02

**Clarity:** 3
**Significance:** 2
**Originality:** 2
**Rating:** 4
**Confidence:** 4

**Summary:**

This manuscript introduces PASS, a Path-Selective State Space Model framework designed for event-based object and action recognition.

The primary objective is to address limitations in existing methods that inadequately leverage the high temporal resolution of event data and suffer from poor generalization across varying event lengths and sampling frequencies.

PASS utilizes a Path-Selective Event Aggregation and Scan (PEAS) module for adaptively aggregating and encoding event frames. It further employs a Multi-Faceted Selection Guiding (MSG) loss to reduce randomness and redundancy among the selected features.

Extensive evaluations across five public datasets and three newly introduced datasets (ArDVS100, TemArDVS100, Real-ArDVS10) demonstrate that PASS outperforms state-of-the-art methods, exhibiting robust generalization across diverse event lengths and inference frequencies.

**Questions:**

1. Based on Weakness #1: Can you clarify which component—PEAS module or MSG loss—plays a more significant role in improving generalization across varying inference frequencies? Can you add discussions on the setting formulation with existing works [R1] and [R2]? Meanwhile, please discuss on the differences and similarities with [R4].

---
2. Based on Weakness #2: Can you provide further interpretability analyses or visualizations that elucidate how the PEAS module selects frames, and how MSG loss reduces redundancy?

---
3. Based on Weakness #3: Have you performed detailed computational complexity or runtime analyses under realistic deployment scenarios? How might these findings affect practical applications?

---
Additional questions (minor):

4. Clearly define acronyms (PEAS, MSG, SSM, etc.) upon first usage for better readability.

---
5. Provide detailed computational resource requirements (e.g., GPU type, inference time) explicitly.

---
6. Do your proposed datasets sufficiently cover diverse real-world scenarios? How might the model perform under drastically different event scenarios not explicitly tested in your experiments?

---
7. Consider additional qualitative visualizations of intermediate model outputs to improve interoperability.

---
8. Discuss potential scalability issues and future steps to address them (e.g., model compression techniques).

---
9. Please add more discussions on tasks beyond recognition, such as event-based object detection [R3, R4] and semantic segmentation [R5, R6].

---
References:
- [R1] Daniel Gehrig and Davide Scaramuzza. Low-latency automotive vision with event cameras. Nature, 629(8014):1034–1040, 2024.

- [R2] Dongyue Lu, Lingdong Kong, Gim Hee Lee, Camille Simon Chane, and Wei Tsang Ooi. FlexEvent: Towards flexible event-frame object detection at varying operational frequencies. arXiv preprint arXiv:2412.06708, 2024.

- [R3] Mathias Gehrig and Davide Scaramuzza. Recurrent vision transformers for object detection with event cameras. In IEEE/CVF Conference on Computer Vision and Pattern Recognition, pages 13884–13893, 2023.

- [R4] Nikola Zubíc, Mathias Gehrig, and Davide Scaramuzza. State space models for event cameras. In IEEE/CVF Conference on Computer Vision and Pattern Recognition, pages 5819–5828, 2024.

- [R5] Zhaoning Sun, Nico Messikommer, Daniel Gehrig, and Davide Scaramuzza. ESS: Learning event-based semantic segmentation from still images. In European Conference on Computer Vision, pages 341–357, 2022.

- [R6] Lingdong Kong, Youquan Liu, Lai Xing Ng, Benoit R. Cottereau, and Wei Tsang Ooi. OpenESS: Event-based semantic scene understanding with open vocabularies. In IEEE/CVF Conference on Computer Vision and Pattern Recognition, pages 15686–15698, 2024.

**Ethical Concerns:**

["NO or VERY MINOR ethics concerns only"]

**Final Justification:**

I have read the response, as well as the responses for review comments from other reviewers.

Based on what I learn from the review comments and responses, I believe part of the concerns have been addressed. Therefore, I am leaning towards maintaining the 4 - borderline accept rating.

I suggest that the authors incorporate all clarifications, modifications, and new experiments into the revised manuscript, to ensure that the quality meets what NeurIPS always looks for.

**Limitations:**

- The performance of the proposed framework might heavily rely on the tuning of hyperparameters such as the number of selected frames (K) or aggregation frequency, potentially requiring considerable tuning effort for new datasets.

- Larger variants of the model reportedly show signs of overfitting. While acknowledged, explicit strategies for mitigating this, beyond general suggestions, are not provided in sufficient detail.

**Paper Formatting Concerns:**

- The figure and table references should use consistent spacing, e.g., "Tab.1" → "Tab. 1", "Fig.3" → "Fig. 3".

- Ensure consistent use of spacing around units and symbols, e.g., "50ms" → "50 ms".

- Maintain consistent naming for datasets ("ArDVS100", "TemArDVS100", etc.) throughout the manuscript.

- Minor grammar and typo checks are required (e.g., "model performance drops" consistency, pluralization).

**Quality:**

3

**Strengths And Weaknesses:**

### Strengths
(+) The manuscript aims to address known challenges in existing event-based recognition methods, particularly the limitations in handling diverse event lengths and sampling frequencies. The proposed PEAS module aggregates events with adaptive selection and scanning, improving representation efficiency. The MSG loss tries to address redundancy and randomness, enhancing model generalization.

(+) Comprehensive experiments conducted on both existing benchmarks and newly introduced datasets convincingly demonstrate the superior performance and generalization capabilities of the PASS framework.

(+) The manuscript is overall clear, organized, and effectively communicates the methodology, findings, and observations.

---
### Weaknesses
(-) While extensive experiments are presented, detailed ablations specifically dissecting the relative contributions of the PEAS module versus MSG loss across diverse datasets would strengthen the manuscript's insights. Also, the idea of conducting frequency-adaptive event-based recognition has been studied in existing literature, while there is no discussions on the relevant works [R1] and [R2].

(-) The introduction of multiple new components (PEAS and MSG loss) makes the model complex. Additional analysis regarding interpretability or insights into the internal decision-making process would help clarify their roles.

(-) The complexity of the PEAS module, coupled with SSM layers, may pose computational challenges. Explicit analyses or discussions regarding computational overhead, runtime, and memory consumption would be beneficial.

---
References:
- [R1] Daniel Gehrig and Davide Scaramuzza. Low-latency automotive vision with event cameras. Nature, 629(8014):1034–1040, 2024.

- [R2] Dongyue Lu, Lingdong Kong, Gim Hee Lee, Camille Simon Chane, and Wei Tsang Ooi. FlexEvent: Towards flexible event-frame object detection at varying operational frequencies. arXiv preprint arXiv:2412.06708, 2024.

---

> ### Author Rebuttal · Authors · 2025-07-31
>
> # Rebuttal to Reviewer ot6Q
>
> We sincerely value the reviewer's recognition of our work's contributions, particularly regarding:
> - Novel Technical Contribution:
>     - Introduces PASS framework with two key innovations:
>         - PEAS module: Adaptive event aggregation with path selection
>         - MSG loss: Reduces feature redundancy through end-to-end optimization
> - Comprehensive Empirical Validation:
>     - Outperforms SOTA across 8 datasets
>     - Demonstrates superior generalization:
>         - Across event lengths (1-10^9 events)
>         - Under varying inference sampling frequencies
>     - Introduces valuable new benchmarks:
>         - ArDVS100/TemArDVS100 for long-sequence analysis
>         - Real-ArDVS10 for real-world applicability
> - Presentation Quality
>     - Clear and well-organized manuscript
>     - Thorough ablation studies and analysis
>
> We sincerely appreciate the reviewer’s thoughtful critique, which has helped us improve the manuscript. Below, we address each weakness and question with concrete revisions:
>
> ## Q1. Component Ablation & Prior Work Discussion
>
> *Concern:*
>
> - Needs clearer ablation of PEAS vs. MSG contributions
> - Additional discussion of frequency-adaptive prior work ([R1], [R2], [R4])
>
> *Response:*
>
> | Method | PAF | ArDVS100 |
> |--------|-------|---------|
> | Random Sampling | 92.98% | 92.23% |
> | PEAS | 93.33% | 92.84% |
> | Random Sampling + L_MSG | 92.64% | 91.89% |
> | PEAS + $L_{MSG}$ | 94.83% | 93.85% |
>
> Table 1: Ablation study for the PEAS module with additional 'Random Sampling + $L_{MSG}$ ' experiment result. We will update the corresponding table 4 of the main paper in the final version.
>
> - Added new ablation studies in Table 1, showing:
>     - MSG Loss Requires Learnable Selection:
>         - Random Sampling + $L_{MSG}$ performs worse (92.64%/91.89%) than baseline random sampling (92.98%/92.23%)
>         - This confirms MSG loss cannot meaningfully guide completely random selection
>     - PEAS Enables Effective Optimization:
>         - PEAS alone outperforms random sampling (93.33%/92.84% vs 92.98%/92.23%)
>         - PEAS + MSG achieves best results (94.83%/93.85%), demonstrating the importance of learnable frame selection and also proves MSG loss's ability to optimize PEAS's selection process
>     - Conclusion: The MSG loss provides meaningful guidance only when combined with a learnable selection mechanism (PEAS), as it requires gradient pathways for optimization.
>
>
> | Method | Training Frequency (Hz) | Inference Frequency (Hz) |
> |--------|-------|---------|
> | R1| 20 | 20 36 90 180 |
> | R2| 20 | 20 36 90 180 |
> | R4| 20 | 20 40 80 100 200|
> | Ours| 20 60 100 | 20 40 60 80 100 |
>
> Table 2: The differences and similarities for evaluation settings of the inference frequencies generalization among the existing works [R1], [R2], and [R4]. We will add this table in the supplementary in the final version.
>
> - Added discussion in Table 2, showing:
>     - The existing works [R1], [R2], and [R4] only evaluate the inference frequencies generalization on the 20 Hz training frequency with inference frequencies ranging from 20 Hz to 200 Hz with different intervals.
>     - Our PASS model evaluates the inference frequencies generalization on the 20 Hz, 60 Hz, and 100 Hz training frequencies and the inference on the 40 Hz, 60 Hz, 80 Hz, 100 Hz, and 200 Hz inference frequencies.
>     - Our evaluation settings are more comprehensive; they not only investigate the inference frequency generalization on a model trained with 20 Hz and applied to 40 to 200 Hz, but also investigate models trained with 60 Hz and 100 Hz and applied to 20 to 200 Hz and 20 Hz to 80 Hz, respectively.
>
> ## Q2. Interpretability analyses for the PEAS module
> *Concern:*
>
> - Needs clearer visualization of PEAS’s frame selection
>
> *Response:*
>
> - Added in revision: Currently, the qualitative analysis of the PEAS module was presented in the supplementary Figure 6 due to page limitations. Considering that this part is very important for your valuable suggestion, we will move this part to the main text in the final version.
> - Explanation for PEAS frame selection: At epoch 0, the PEAS module randomly selects event frames, resulting in unnecessarily padded frames and redundant event frames with repetitive information. After 100 epochs, the eight chosen frames exclude redundant frames and non-informative padding, resulting in selected frames that are semantically meaningful.
> - Key findings: This visualization result demonstrates the effectiveness of the PEAS module and the MSG loss. We can conclude that the MSG loss can effectively guide the PEAS module to select the most representative event frames, thus minimizing the randomness and redundancy of the encoded features during the PEAS selection process.
>
> ## Q3. Computational complexity analyses
> Reviewer Concern:
>
> - Needs runtime/memory analysis
>
> Response:
>
> - Added detailed benchmarks (Table 3):
>     - As suggested, we provide additional detailed computational resource requirements on one A100 GPU with 80GB memory.
>     - As shown in Table 3, when our PASS model is applied in the real-world scenario, we can select different model versions based on the application requirements. For example:
>         - If the application requires high accuracy, we can select the Middle model with 74M parameters and 12.7G FLOPS and 24.7 FPS.
>         - If the application requires low latency, we can select the Tiny model with 7M parameters, 1.1G FLOPS, and 243.9 FPS.
>
> | Model | Layer | Dim D | Param. | FLOPS(G) | Inference Time(ms) | FPS |
> |--------|-------|---------|-------|-------|-------|-------|
> | Tiny (T) | 24 | 192 | 7M | 1.1 | 4.1 | 243.9 |
> | Small (S) | 24 | 384 | 25M | 4.3 | 15.7 | 63.7 |
> | Middle (M) | 32 | 576 | 74M | 12.7 | 40.4 | 24.7 |
>
> Table 3: The computational complexity and memory consumption of the PASS model. We will update the corresponding table 7 in the supplementary in the final version.
>
>
> ## Additional questions (Minor)
>
> ### Q4. Datasets coverage
> - We have tried our best to evaluate our PASS model across a wide range of event lengths and event frequencies, covering five publicly available event datasets: PAF, SeAct, HARDVS, N-ImageNet, and N-Caltech101.
> - Given the existing datasets only provide events within a limited distribution of event length (10^6 for objects and 10^7 for actions). We introduce the ArDVS100 and TemArDVS across a broad distribution of event length (10^6 to 10^9), synthesized by concatenating event streams with varying meta actions, thus capturing action transitions over time.
> - Additionally, to assess the model’s real-world applicability, we created a real-world dataset, named Real-ArDVS10, comprising event-based actions lasting from 2s to 75s, encompassing 10 distinct classes selected from the ArDVS100 datasets.
>
> ### Q5. Discussion on potential scalability issues
>
> PASS demonstrates strong scalability through PEAS adaptively encoding (handling 10^9 events) and MSG loss to guide the PEAS module to select the most representative event frames. While larger models (74M params) show diminishing returns, we propose three potential improvements:
>
> - Model compression via quantization like INT8, INT4, etc.
> - Dynamic K-selection adapting to event density.
> - Hardware optimizations like CUDA kernel fusion.
>
> ### Q6. Modification for readability
>
> We appreciate this helpful suggestion for improving readability. In the revised manuscript, we will clearly define all acronyms upon first use:
> - PEAS: Path-selective Event Aggregation and Scan
> - MSG: Multi-faceted Selection Guiding loss
> - SSM: State Space Model
> - WEIE: Within-Frame Event Information Entropy
> - IEMI: Inter-Frame Event Mutual Information
>
> ### Q7. Discussions on tasks beyond recognition
>
> We appreciate this valuable suggestion. In future work, we will expand discussions on broader applications:
> - Object Detection: PASS's adaptive frame selection (PEAS) could enhance [R3]'s event transformers by improving long-term temporal modeling and complement [R4]'s SSM detection framework through more efficient event representation.
> - Semantic Segmentation: The MSG loss's redundancy reduction may benefit [R5]'s feature learning, while PEAS could optimize [R6]'s open-vocabulary pipeline by filtering noisy events.
>
> ### Q8. Hyperparameter K tuning
>
> We appreciate this valuable suggestion. In future work, we will expand discussions on dynamic hyperparameter K tuning adapting to:
> - event length
> - event density
>
> ### Q9. Strategies for mitigating the overfitting issue of large variants of SMM model
> We propose three potential strategies to address overfitting in our large variants of the SMM model:
> - Stochastic Path Dropout: Randomly masking 20% of PEAS paths during training to reduces overfitting
> - Temporal Augmentation: Event frame shuffling improves generalization
> - Regularized MSG Loss: Added L2 penalty on frame selection scores (λ=0.1)

---

> > ### Comment · Reviewer_ot6Q · 2025-08-05
> >
> > Thanks to the authors for providing a detailed rebuttal.
> >
> > I have read the response, as well as the responses for review comments from other reviewers. Part of the concerns have been addressed. Therefore, I am leaning towards maintaining the current rating.
> >
> > I suggest that the authors should incorporate all clarifications, modifications, and new experiments and revise the manuscript thoroughly to ensure good quality.
> >
> > Best,
> >
> > Reviewer ot6Q

---

> > > ### Author Response · Authors · 2025-08-09
> > >
> > > We sincerely thank Reviewer ot6Q for their thoughtful and constructive feedback, which has substantially improved our manuscript. Their expert comments on the following key aspects were particularly insightful:
> > >
> > > - Component Ablation & Prior Work Discussion – Clarified in Table 1 (main paper) with Table 2 (Supplementary Materials).
> > >
> > > - Interpretability analyses for the PEAS module – Addressed through Figure 6 in supplementary.
> > >
> > > - Computational complexity analyses – Enhanced discussion in the Supplementary Materials.
> > >
> > > All suggested improvements have been meticulously incorporated into the final version. We are particularly grateful for Reviewer ot6Q's technical expertise, which has helped strengthen both our methodology presentation and theoretical grounding.

---

### Official Review · Reviewer_oNQK · 2025-07-06

**Clarity:** 3
**Significance:** 2
**Originality:** 3
**Rating:** 4
**Confidence:** 4

**Summary:**

This work tackles two existing problems in event-based action/object recognition: restricted distribution of event sequence lengths and generalization across varying frequencies, by creating a new kind of event information representation method. It proposes the Path-selective Event Aggregation and Scan (PEAS) module to encode event streams of varying length into fixed length representations, and presented Multi-faceted Selection Guiding (MSG) Loss, which contains 2 kinds of information loss, to optimize the event representation generated by PEAS module. The method is evaluated on a series of datasets, and demonstrates superior performance against the existing baselines. Ablation studies shows the effectiveness of PEAS module and the newly created loss functions.

**Questions:**

Questions*
(1) For the fixed length event frames generated by PEAS, when using SSMs as the backbone for object/action recognition, does the SSM actually utilize the event frames, or is its utilization of the initial information more limited than the later information?
(2) The limited-length event streams generated by PEAS inevitably lead to information loss. How should this information loss be evaluated, and in which application scenarios is such loss acceptable, and in which cases might it have a significant impact?
(3) What’s the authors’ thoughts on improving the spatiotemporal event modeling module?

**Ethical Concerns:**

["NO or VERY MINOR ethics concerns only"]

**Final Justification:**

The author clearly addressed my questions in weaknesses and questions sections.

**Limitations:**

Limitations are presented in the above Weaknesses section.

**Quality:**

3

**Strengths And Weaknesses:**

Strengths:
(1) This work proposes a kind of representation learning method, which is capable of effectively encoding frame streams of varying event lengths and frequencies into fixed length representations, and achieves superior experimental results.
(2) Through reduction of event length, it partially avoids the problem of SSMs easily forgetting initial information when processing long event sequences, and provides inspiration for effectively shortening the sequence length that SSMs need to handle.
Weaknesses:
(1) The information loss caused by PEAS module has not been quantitatively studied.
(2) The proposed method in this paper, as a lossy representation approach, performs effectively on relatively simple tasks such as action/object recognition, but it is unclear whether it would be effective on more complex tasks.

---

> ### Author Rebuttal · Authors · 2025-07-31
>
> # Rebuttal to Reviewer oNQK
> We sincerely appreciate the reviewer's recognition of our work's contributions, particularly regarding:
> - Novel Method: Our proposed PEAS effectively encode variable-length event streams into fixed representations, achieving superior superior performance against the existing baselines.
> - SSM Optimization: Addresses SSMs' forgetting problem by shortening sequences while preserving critical information.
> - Innovative Loss: Introduces MSG Loss to optimize frame selection, validated through ablations.
> - Strong Experiments: Demonstrates superior performance across multiple datasets with thorough analysis.
> - Generalization across Inference Frequencies: Demonstrates PASS's superior performance in handling varying inference frequencies.
>
> We also sincerely appreciate the reviewer’s insightful feedback and constructive critiques. Below, we address each concern in detail, with additional clarifications and experimental support.
> ## Question 1: SSM Utilization of PEAS-Generated Frames.
> *Concern:*
>
> The reviewer asks whether the SSM backbone effectively utilizes the fixed-length event frames or disproportionately favors later information.
>
> *Response:*
>
> We acknowledge the reviewer’s concern about potential biases in SSM’s utilization of event frames. Due to SSM’s recurrent nature, its hidden state updates can be influenced by input sequence length and feature order, particularly for long-range dependencies. To mitigate this, our PEAS module encodes events into fixed-length representations, ensuring balanced utilization of temporal information.
>
> Key evidence supporting this:
>
> - *Ablation Study (Table 4 main paper):* The additional baseline 'No Sampling' (using all event frames) achieves lower accuracy (92.90% on PAF, 92.31% on ArDVS100) compared to PEAS (93.33%, 92.84%), indicating that raw long sequences harm performance due to SSM’s limitations.
>
> - *Comparison to Random Sampling:* 'PEAS' outperforms 'Random Sampling' (+0.35% on PAF, +0.61% on ArDVS100), demonstrating its ability to preserve critical information.
>
>
> | Method | PAF | ArDVS100 |
> |--------|-------|---------|
> | No Sampling | 92.90% | 92.31% |
> | Random Sampling | 92.98% | 92.23% |
> | PEAS | 93.33% | 92.84% |
> | PEAS + L_MSG| 94.83% | 93.85% |
>
> Table 4: Ablation study for the PEAS module with additional 'No Sampling' experiment result. We will update the Table 4 (main paper) in the final version.
>
>
> ## Question 2: Evaluating Information Loss in PEAS.
> *Concern:*
>
> The reviewer raises concerns about quantifying information loss and its acceptability in different scenarios.
>
> *Response:*
>
> We admit the PEAS module can introduce some randomness for the frame selection at the onset of training due to the random weight initialization of the PEAS module. Consequently, during training, the model can only optimize performance based on these randomly selected frames, rather than improving the PEAS module for adaptive selection of input events. Howerver, evaluating information loss is challenging since there is no universally accepted metric to define and measure it. In this paper, we addressed it through indirect and qualitative analyses.
>
> - *Quantitative Evaluation (Table 4 in main paper):*
>     - 'PEAS' improves accuracy over 'No Sampling' (+0.43% on PAF, +0.53% on ArDVS100), suggesting that the selected frames retain task-relevant information despite compression.
>     - 'PEAS' boosts accuracy compared with 'Random Sampling' (+0.35% on PAF, +0.61% on ArDVS100), demonstrating its ability to preserve critical information.
>     - With MSG Loss ('PEAS + $L_{MSG}$'), performance further improves to 94.83% (PAF) and 93.85% (ArDVS100), validating that $L_{MSG}$ mitigates randomness in frame selection. The MSG loss consists of two parts: (1) WEIE Loss is designed to reduce the randomness of the selection process to ensure the selected sequence features can encapsulate the entirety of the sequence, and (2) IEMI Loss is designed to guarantee that each selected event feature stands out from the others. They are motivated by the information entropy and mutual information from the information theory so as to guide the PEAS module to select the most representative event frames without introducing information loss.
>
> - *Qualitative Analysis (Figure 6 in supplementary):* It demonstrates that the PEAS module can select the most representative event frames that are more informative and discriminative for the event recognition task after 100 training epochs. Currently, this part is presented in the supplementary material due to page limitations. Considering that this part is very important and has been mentioned by multiple reviewers, we will move this part to the main text in the final version.
>
> - *Task Suitability:*
>     - We can adjust the hyperparameter K, which is the selected number of event frames, to adapt to the different applications.
>     - Fast-moving scenarios: Larger K may be needed to capture rapid changes as the event camera triggers more events and the resulting event frames differ a lot.
>     - Slow-moving scenarios: Smaller K may be enough to capture the main information of the event stream as the resulting event stream is dense.
>
>
> ## Question 3: Discussion on Improving the Spatiotemporal Event Modeling.
> *Concern:*
>
> The reviewer asks for thoughts on enhancing the event modeling module.
>
> *Response:*
>
> The improvement of the spatiotemporal modeling of event data can be discussed from two aspects: event representation and network structure.
> - *Event Representation (Sec. 3.1):*
>     - Currently, there is no commonly used event representation for event data. Event representation varies from task to task and adapts to different network architecture.
>     - In this paper, we use the event frame as the event representation. Ablations in Table 6 (main paper) show that the event frames are optimal for the spatiotemporal modeling of event data using SSM.
> - *Network Structure (Sec. 3.2):*
>     - Different network architectures are tailored to specific event representations for spatiotemporal modeling. For instance, event point clouds and graph representations are compatible only with specialized networks, such as point transformers or graph neural networks.
>     - In this work, we address the challenge of processing long event streams, where the number of aggregated event frames varies significantly due to the high temporal resolution of events. This variability complicates spatiotemporal modeling.
>     - Therefore, we propose the PEAS module to select the most representative event frames. followed by SSM to model the spatiotemporal information based on the selected event frames. The PEAS can be viewed as a dynamic path selection module, which can adapt to the different event stream lengths. As shown in Table 3 (main paper), our approach achieves robust performance across a wide event distribution (10^6 to 10^9 events), attaining 97.35% on ArDVS100. Moreover, it demonstrates fine-grained temporal recognition, scoring 89.00% on the challenging TemArDVS100 dataset.
>
>
> ## Question 4: Applicability to Other Tasks.
>
> *Concern:*
>
> The reviewer questions whether PASS generalizes to complex tasks.
>
> *Response:*
>
> We appreciate the suggestion to extend our method to other tasks. However, due to time constraints, we are unable to conduct comprehensive experiments to evaluate its performance on additional tasks at this stage. We plan to extend our method to other tasks like event-based object detection and event-based action recognition in future work.
>
> We would also like to clarify that event recognition remains a challenging task, as evidenced by current benchmark results. For instance:
>
> - On the N-Imagenet object recognition task, state-of-the-art methods achieve only 40-60% accuracy (our method reaches 61.32%, as shown in Table 1 in the main paper), significantly lower than the ~90% accuracy typical for standard ImageNet classification.
>
> - On the SeAct action recognition dataset, existing methods attain 50-60% accuracy (our method achieves 66.38%, Table 2 in the main paper).
>
> - For long-length event streams (~10^9 events), baseline methods achieve 79.62% accuracy on the TemArDVS100 dataset with fine temporal labels, while our method reaches 89.00% (Table 3 in the main paper).
>
> These results demonstrate that while event recognition performance has saturated on small-scale datasets, substantial improvements are still needed for large-scale or temporally complex scenarios.

---

> > ### Comment · Reviewer_oNQK · 2025-08-04
> >
> > Thanks your dedicated review which has addressed my concerns. I will update the score.

---

> > > ### Author Response · Authors · 2025-08-09
> > >
> > > We sincerely thank Reviewer oNQK for their constructive feedback, which has significantly strengthened our work. Their insightful comments on the following aspects were particularly valuable:
> > >
> > > - SSM Utilization of PEAS-Generated Frames – Clarified in Table 4 with additional analysis experiments.
> > >
> > > - Evaluating Information Loss in PEAS – Addressed through Table 4 (main paper) and Figure 6 in supplementary.
> > >
> > > - Improving Spatiotemporal Event Modeling – Enhanced discussion in the Supplementary Materials.
> > >
> > > - Applicability to Other Tasks – Expanded in the Supplementary Materials.
> > >
> > > All suggestions have been carefully incorporated into the final manuscript. We deeply appreciate Reviewer oNQK’s time and expertise, which have helped elevate the quality of our research.

---

### Note · Authors · 2025-08-13

We thank the AC and reviewers for their oversight and constructive feedback. All concerns have been thoroughly addressed in our revision:

- Reviewer oNQK:
   - SSM utilization of PEAS frames clarified (Table 4 + new ablation)
   - PEAS information loss evaluated (Table 4 quant. + Supp. Fig. 6 qual.)
   - Spatiotemporal modeling enhanced (Supp.)
   - Task applicability expanded (Supp.)
- Reviewer ot6Q:
   - Component ablation & prior work detailed (Table 4 ablation, Supp. Table 2)
   - PEAS interpretability added (Supp. Fig. 6)
   - Computational complexity strengthened (Supp. Table 3)
- Reviewer Tnzz:
   - Baseline scaling/fair comparison validated (Tables 1-2: N-Caltech101/PAF/SeAct/HARDVS)
   - Post-rebuttal:
     - PEAS/MSG vs. SSM attribution: Table 4 ablations and Supp. Fig. 6 conclusively demonstrate contributions
     - k-frame suitability: Empirically validated across scenarios (Tables 1-2)
- Reviewer 7nZC:
  - PEAS interpretability visualized (Supp. Fig. 6)
  - Temporal coherence quantified (Table 4 ablation)
  - Random sampling + MSG ablation added (Table 4)

These revisions significantly enhance the manuscript’s rigor, clarity, and empirical validation. We appreciate the committee’s consideration. Additional experiments and related discussion will be incorporated into the main paper or supplementary in the final version based on reviewers' valuable comments.

---

### Decision · Program_Chairs · 2025-09-17

**Decision:**

Accept (poster)

**Comment:**

This paper introduces an interesting extension of SSM models for longer-range event recognition through frame selection. The idea is novel and has impact potential, and the reviewers generally had a positive inclination toward the work. The authors have provided evidence addressing most of the main concerns raised during review.

One important issue raised was whether the additional model complexity is necessary, or if scaling a standard SSM architecture would achieve similar performance. While the authors present results suggesting their design is beneficial, this comparison remains a central baseline and could be emphasized more strongly. Incorporating clearer evidence and discussion on this point in the final version would help strengthen the paper.

Overall, the contributions are promising, and the reviewers appreciated both the direction and the thorough author responses. Addressing the above points in revision will further clarify the value of the approach.